# Nonpathogenic Pseudomonas syringae derivatives and its metabolites trigger the plant "cry for help" response to assemble disease suppressing and growth promoting rhizomicrobiome

Yunpeng Liu [1,4], Huihui Zhang[2,3,4], Jing Wang[2], Wenting Gao[1], Xiting Sun[1], Qin Xiong[1], Xia Shu[1], Youzhi Miao[2], Qirong Shen [2], Weibing Xun [2] ✉ & Ruifu Zhang [1,2] ✉

Plants are capable of assembling beneficial rhizomicrobiomes through a "cry for help" mechanism upon pathogen infestation; however, it remains unknown whether we can use nonpathogenic strains to induce plants to assemble a rhizomicrobiome against pathogen invasion. Here, we used a series of derivatives of *Pseudomonas syringae* pv. *tomato* DC3000 to elicit different levels of the immune response to *Arabidopsis* and revealed that two nonpathogenic DC3000 derivatives induced the beneficial soil-borne legacy, demonstrating a similar "cry for help" triggering effect as the wild-type DC3000. In addition, an increase in the abundance of *Devosia* in the rhizosphere induced by the decreased root exudation of myristic acid was confirmed to be responsible for growth promotion and disease suppression of the soil-borne legacy. Furthermore, the "cry for help" response could be induced by heat-killed DC3000 and flg22 and blocked by an effector triggered immunity (ETI) -eliciting derivative of DC3000. In conclusion, we demonstrate the potential of nonpathogenic bacteria and bacterial elicitors to promote the generation of disease-suppressive soils.

The rhizosphere microbiome plays a critical role in modulating plant growth and health[1]. Some rhizobacteria closely interact with roots and exert diverse plant-beneficial functions. These plant beneficial rhizobacteria generally protect plants from disease by directly antagonizing soil-borne pathogens or inducing plant resistance[2,3]. A disease-suppressive rhizosphere microbiome can be formed by enriching these beneficial rhizobacteria, thereby performing consistent and long-lasting protection for the plant[1].

Beneficial rhizosphere microbes can be recruited by plants upon pathogen attack in a process termed "cry for help" response[4–8]. The recruited and assembled beneficial rhizosphere microbiome will further result in the formation of long-lasting disease suppressive soil[9].

[1]State Key Laboratory of Efficient Utilization of Arid and Semi-arid Arable Land in Northern China, Institute of Agricultural Resources and Regional Planning, Chinese Academy of Agricultural Sciences, Beijing 100081, P. R. China. [2]Jiangsu Provincial Key Lab for Organic Solid Waste Utilization, Jiangsu Collaborative Innovation Center for Solid Organic Waste Resource Utilization, Nanjing Agricultural University, Nanjing 210095, P. R. China. [3]Department of Agronomy and Horticulture, Jiangsu Vocational College of Agriculture and Forestry, Zhenjiang, Jiangsu 212400, P. R. China. [4]These authors contributed equally: Yunpeng Liu, Huihui Zhang. ✉e-mail: xunwb@njau.edu.cn; rfzhang@njau.edu.cn

The "cry for help" response strategy is a potential pathway for generating a disease suppressive rhizosphere microbiome that is well recognized in several plant–pathogen interaction models, such as *Arabidopsis-Pseudomonas syringae*, *Arabidopsis-Hyaloperonospora arabidopsidis*, bean-*Fusarium oxysporum* and sugar beet-*Rhizoctonia solani*[4,6,8,10]. However, due to the virulence of these pathogens to crops, they are not applicable for agricultural production. Moreover, the mechanism whereby the plant "cry for help" response is induced by pathogen attack is unknown. These limitations urge the exploration of harmless elicitors to trigger the plant "cry for help" response, which will largely facilitate sustainable agricultural development.

Upon infecting plants, pathogens release a variety of molecules that regulate plant immunity and finally alter the composition of root exudates[11–14]. Such altered root exudation further shapes the rhizosphere microbiome[15,16]. These changes are initiated by the recognition of pathogens by plant cells. The recognition process includes general sensing of microbe-associated molecular patterns (MAMPs) by plant pattern recognition receptors (PRRs) and the recognition of specific effector proteins by nucleotide-binding leucine-rich repeat (NLR) immune receptors[17,18]. The recognition of MAMPs elicits pattern-triggered immunity (PTI) that restricts further infection[19]. Pathogen-secreted effectors can disarm plant PTI, but additional recognition of the effectors elicits a stronger immune response known as effector-triggered immunity (ETI)[20]. Pathogens that are able to successfully infect plants are always capable of bypassing recognition by plants. For instance, the well-studied model pathogen *Pseudomonas syringae* pv. *tomato* (*Pst*) DC3000 has a conserved peptide derived from flagellin, flg22, which is sensed by plants to elicit PTI. DC3000 secretes 36 effectors through the type III secretion system (T3SS); some of these effectors are capable of disabling PTI[21], while some of these effectors are recognized by plant cells to elicit ETI, such as HopM1 and HopQ1-1, which are recognized by *Nicotiana benthamiana*[22,23]. These interactions between pathogens and the plant immune response might be one of the elicitors of the "cry for help" response. Pathogens also produce phytohormone analogs that interfere with plant defense signaling[18]. In addition, many unknown metabolites produced by phytopathogens affect plant signaling. These interactions finally lead to alterations in the root exudate composition[7,12], which functions as a direct factor mediating the recruitment of beneficial rhizosphere microbes[24]. However, it remains unclear which process in the plant–pathogen interaction triggers the "cry-for-help" response. Can non-pathogenic phytopathogens be used to induce plants to assemble a specific rhizosphere microbiome to protect plants given that virulent pathogens are harmful and inapplicable for agricultural production?

In this study, we explored the effect of different derivatives of *Pst* DC3000, a successful pathogen of *Arabidopsis* and tomato, on the composition of the rhizosphere microbiome. We demonstrate that two nonpathogenic derivatives of *Pst* DC3000 can induce plants to establish a disease suppressive soil-borne legacy. We further revealed that the "cry for help" response could be induced by the metabolites of the nonpathogenic derivatives and blocked by the ETI-eliciting derivative of *Pst* DC3000.

## Results

### Eliciting different immune responses of *Arabidopsis* by *Pst* DC3000 derivatives
Several nonpathogenic derivatives of *Pst* DC3000 were deployed to induce different levels of immune responses in Arabidopsis. *Pst* DC3000 uses T3SS to deliver type III effectors (T3Es) into plant cells to inhibit the immune response[25,26]. The *Pst* DC3000-derived nonpathogenic mutant D36E, in which the T3SS remains intact but all 36 type III effectors are deleted[21], was used as a strong PTI-eliciting strain (Fig. 1A). The nonpathogenic derivative D36EFLC was obtained by deleting the flagellin coding gene *fliC* from D36E and was used as a weak PTI-eliciting strain[23] (Fig. 1A). D36EHPM is another

nonpathogenic strain generated by introducing the coding sequence of HopM1 in D36E as an ETI-eliciting strain (Fig. 1A)[23,27]. We then checked the virulence of the strains by *Arabidopsis* leaf infiltration, and found that only wild-type *Pst* DC3000 (WT) caused disease symptoms, while all other derivatives (D36E, D36EFLC and D36EHPM) did not (Fig. 1B). A hyposensitive response was observed in D36EHPM-infiltrated leaves (Fig. 1B). All the nonpathogenic derivatives showed significantly (two-sided ANOVA, $\alpha = 0.05$, $n = 8$ biologically independent samples, $P < 0.001$) reduced infection than WT D3000 did in Arabidopsis (Fig. 1C). The salicylic acid (SA)-signaling pathway, and especially the expression of *PR1* in *Arabidopsis*, is a defense response marker that is highly activated upon attack by *Pst* DC3000[28]. The induction of the SA signaling pathway was significantly (two-sided ANOVA, $\alpha = 0.05$, $n = 6$ biologically independent samples $P < 0.001$) weaker in the D36EFLC inoculation treatment, indicating a lower level of defense response activation (Supplementary Fig. 1). Both D36E and D36EHPM induced the expression of genes involved in the SA signaling pathway but reduced that of genes in the jasmonic acid (JA) signaling pathway (Supplementary Fig. 1), possibly due to the antagonism between the SA and JA pathways[29]. D36E and D36EHPM triggered an ROS burst in *Arabidopsis* leaves, which is a signature of the immune response, whereas D36EFLC triggered only a very weak ROS burst (Fig. 1D, E). Based on the above results, different levels of plant immune response can be activated by a series of *Pst* DC3000 derivatives, including a pathogenic wild-type strain (WT) inducing weak PTI, a nonpathogenic strain inducing weak PTI (D36EFLC), a nonpathogenic strain inducing strong PTI (D36E), and a nonpathogenic strain inducing ETI (D36EHPM) (Fig. 1F).

### Nonpathogenic DC3000 derivatives drive the alteration of root exudates
Due to the essential role of root exudates in shaping the rhizomicrobiome, we inoculated leaves of 4-week-old gnotobiotic Arabidopsis grown in MS medium with *Pst* DC3000 and the derivatives and analyzed the composition of root exudates by GC–MS. The 97 compounds identified were classified into 9 groups, including amino acids, esters, ketones, primary organic acids, long chain organic acids, sterols, sugars, sugar alcohols and sugar acids (Fig. 2A, Supplementary data 1). PCoA showed that inoculation with any of the DC3000 derivatives altered the composition of root exudates (Fig. 2B). Treatment with either DC3000 or the derivatives increased the relative contents of long chain organic acids (LCOAs) (two-sided ANOVA, $\alpha = 0.05$, $n = 6$ biologically independent samples, $P < 0.001$) and amino acids (two-sided ANOVA, $\alpha = 0.05$, $n = 6$ biologically independent samples, $P < 0.001$) in root exudates (Fig. 2C, D). The relative contents of most of the detected LCOAs and amino acids were increased under either treatment of the DC3000 derivatives (Fig. 2D, E). Since LCOAs and amino acids have been reported to be a group of key compounds in root exudates that are responsible for recruiting beneficial rhizobacteria by Arabidopsis upon infection by DC3000[5], we proposed that the nonpathogenic derivatives of DC3000 might induce the assembly of a beneficial rhizosphere microbiome similar to that induced by WT DC3000.

### The soil-borne legacy of plants treated with nonpathogenic DC3000 derivatives is disease suppressive and growth promoting
As the wild-type *Pst* DC3000-induced soil-borne legacy has been proven to be disease-suppressive[5], we wondered if the soil-borne legacy induced by its nonpathogenic derivatives would be disease-suppressive as well. Then, we applied three generations of enrichment of soil-borne legacy with plants treated with these strains in soil conditions (Fig. 3A, Supplementary Fig. 2, Supplementary Table 1). In brief, one generation included 7-day-old Arabidopsis plants that were transferred into pots, allowed to grow for 20 days, inoculated with these strains, respectively, grown for another 7 days, and removed.

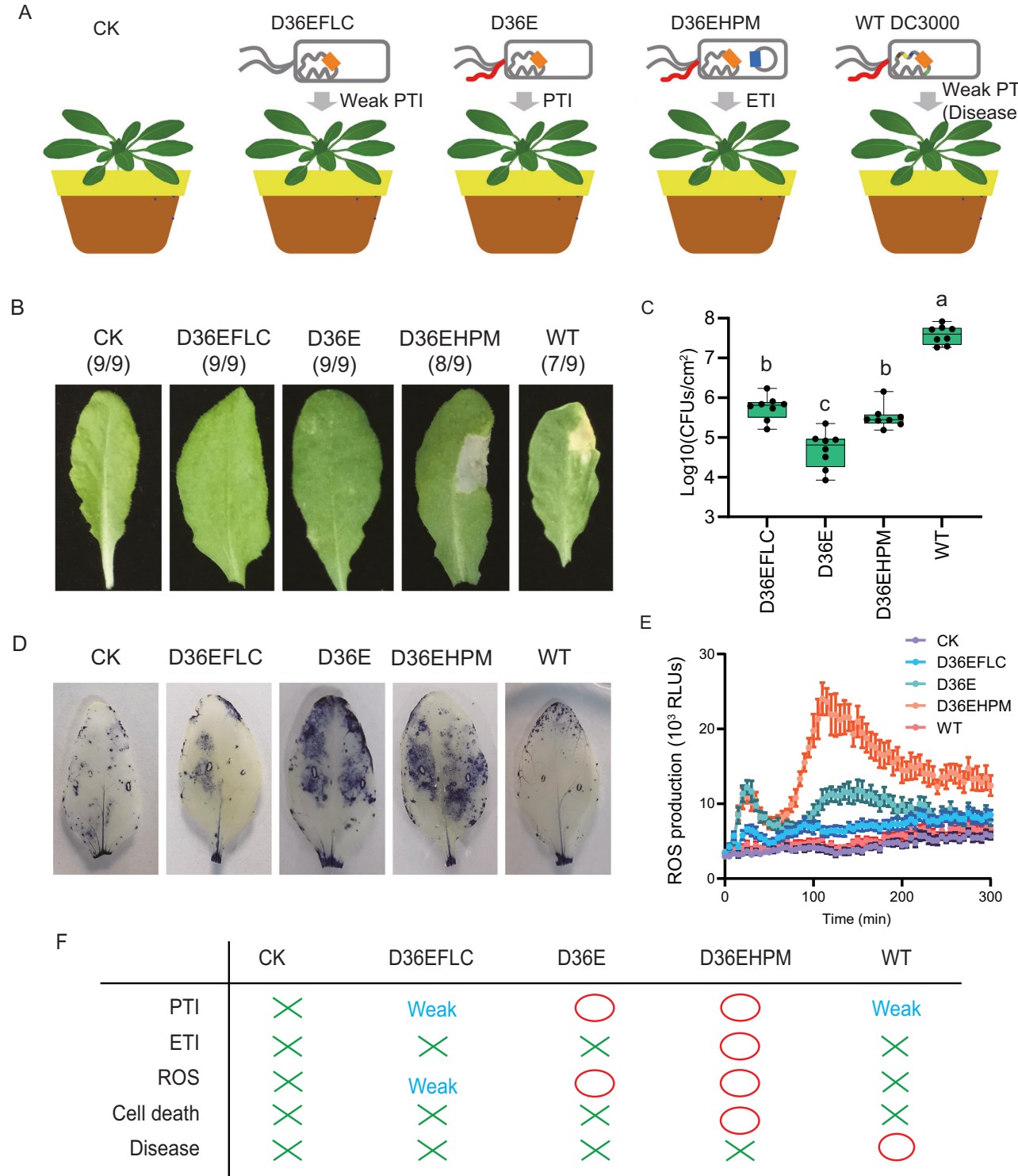

**Fig. 1 | Arabidopsis immune response triggered by DC3000 derivatives.**
**A** Selection of strains used for triggering Arabidopsis immune responses.
**B** Phenotype of leaves infiltrated with the bacteria. Leaves of 4-week-old Arabidopsis grown in vermiculite:peat were infiltrated with wild-type *Pst* DC3000 or the derivatives. The bacterial cells were suspended to a final concentration of $10^6$ cells/mL. Nine replicates were included for each treatment. The fraction under each image indicates the number of times that similar symptoms were observed relative to the number of test inoculations. **C** Bacterial densities in Arabidopsis leaves. Four-week-old *Arabidopsis* was infiltrated with suspensions of wild-type *Pst* DC3000 or the derivatives at $10^5$ cells/mL. Bacteria densities was measured at 3 days post-inoculation using. Boxplot: median, 25%/75% percentiles, and the highest, lowest,

and extremely values are shown. The different letters indicate significant differences according to two-sided ANOVA based on Duncan's multiple range test ($\alpha = 0.05$, $n = 8$ biologically independent samples, $P < 0.001$). **D** NBT staining of $O_2^-$ in leaves of 4-week-old *Arabidopsis* grown in vermiculite:peat. DC3000. NBT staining was performed at 15 h post-inoculation ($n = 6$ biologically independent samples). **E** Arabidopsis leaves infiltrated with DC3000 or derivatives thereof were immediately assayed for ROS production using L-012 chemiluminescence. Data are presented as the mean values +/- SEMs ($n = 6$ biologically independent samples). **F** Overview of the plant response to bacterial strains. "X" indicates no induction by the strain, "O" indicates induction by the strain, and "weak" indicates weak induction by the strain.

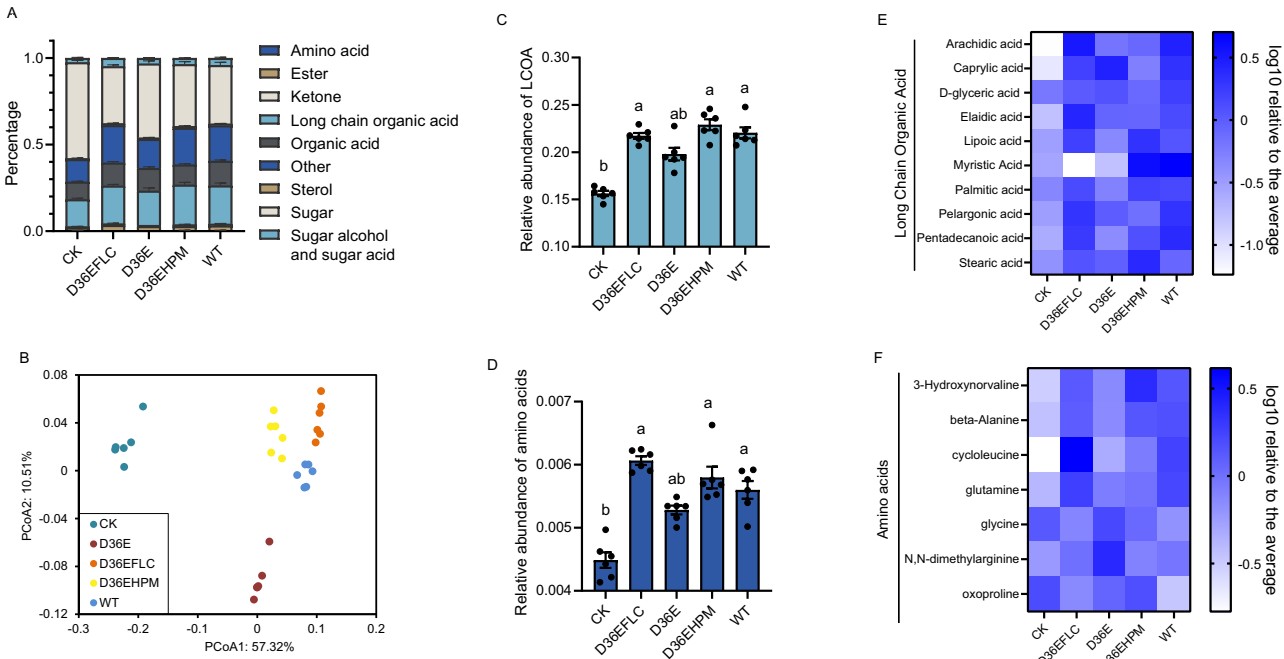

**Fig. 2 | Profile of root exudates in response to *Pst* DC3000-derived strains.**
**A** Relative abundance of grouped root exudate compounds. Data are presented as the mean values +/- SEMs ($n = 6$ biologically independent samples). **B** Principal coordinate analysis (PCoA) of the composition of root exudates composition between untreated control plants, wild-type DC3000- and the DC3000 derivative-treated plants based on Bray–Curtis dissimilarity. **C, D** Relative total abundance of long chain organic acids (LCOAs) (**C**) and amino acids (**D**) in root exudates. The peak area values were normalized and transformed by the log-centered method. Data are presented as the mean values of the log10 relative content to the average +/- SEMs. The different letters indicate significant differences according to two-sided ANOVA based on Duncan's multiple range test ($\alpha = 0.05$, $n = 6$ biologically independent samples, $P < 0.001$). Data are presented as the mean values +/- SEMs. **E, F** Relative abundance of each compound of long chain organic acids (LCOAs) (**E**) and amino acids (**F**) in root exudates. The peak area values were normalized and transformed by the log-centered method. Data are presented as the mean values of the log10 contents relative to the average. A darker blue color indicates a higher value.

After three generations, we then sowed five Arabidopsis seeds into each pot with soil containing the legacy of D36EFLC-, D36E-, D36EHPM-, WT-infected or uninfected Arabidopsis (hereafter referred to as SL-D36EFLC, SL-D36E, SL-D36EHPM, SL-WT and SL-CK, respectively). Germination ratios and plant fresh weights were recorded at 10 days post-sowing and 5 weeks post-sowing, respectively. The results showed that the plants grown in SL-D36EFLC and SL-D36E have a germination ratio (two-sided ANOVA, $\alpha = 0.05$, $n = 20$ independent pots, $P < 0.001$) and fresh weight (two-sided ANOVA, $\alpha = 0.05$, $n = 34$, 70, 71, 53, 33 independent plant seedlings for each box, $P < 0.001$) that were more than twofold higher than those of plants grown in SL-CK, while the plants grown in SL-WT and SL-D36EHPM showed no significant difference from those grown in SL-CK (Fig. 3B, C and D). This is consistent with the previous finding that the wild-type *Pst* DC3000-induced rhizosphere microbiome did not affect plant growth[5]. At 5 weeks post-sowing, we infiltrated the leaves with *Pst* DC3000 and evaluated the disease suppression of the soil legacy at 7 days post-infiltration. SL-WT conferred stronger resistance (two-sided ANOVA, $\alpha = 0.05$, $n = 20$ biologically independent infection experiments, $P < 0.001$) against DC3000 in plants (Fig. 3E). Importantly, SL-D36EFLC and SL-D36E enhanced the resistance of plants against DC3000, as did SL-WT (Fig. 3E). Arabidopsis grown in SL-CK and SL-D36EHPM both showed disease symptoms. These findings indicate that the non-pathogenic derivatives may serve as less-harmful elicitors to induce disease suppression and plant growth-promoting soil legacy (Fig. 3F).

## Composition of the rhizosphere microbiome of plants grown with soil legacy

We next investigated how the rhizosphere microbiome was assembled. The composition of the rhizosphere bacterial communities from the plants grown in legacy-containing soil after the leaf infiltration of wild-type DC3000 was analyzed by amplicon sequencing. We searched the 16 S rDNA sequence of DC3000 against all the generated reads and found no significant hits, indicating no DC3000 contamination in the soil. The alpha diversity indices were highest in the rhizosphere bacterial community of D36E (two-sided ANOVA, $\alpha = 0.05$, $n = 6$ independent samples, $P = 0.001$ for Shannon, $P = 0.018$ for Chao1) and lower in the other treatments (Supplementary Fig. 3A, B). The PCoA results revealed that the bacterial communities under the D36E and D36EFLC treatments were significantly different from those under the other treatments (ANOSIM R = 0.327, $p < 0.001$), which could be distinguished on the horizontal axis (Fig. 4A). The bacterial phyla Proteobacteria (32.1%–38.3%) and Actinobacteria (15.4%–20.7%) were the most abundant groups in all of the communities (Supplementary Fig. 3C). At the genus level, *Sporichthya* (two-sided ANOVA based on Kruskal−Wallis test, $\alpha = 0.05$, $n = 6$ independent samples, $P < 0.001$) and *Actinophytocola* (two-sided ANOVA based on Kruskal−Wallis test, $\alpha = 0.05$, $n = 6$ independent samples, $P < 0.001$), which belong to the phylum Actinobacteria, and *Devosia* (two-sided ANOVA based on Kruskal−Wallis test, $\alpha = 0.05$, $n = 6$ independent samples, $P < 0.001$), which belongs to the phylum Proteobacteria, were differentially distributed in each treatment (Supplementary Fig. 3D−F). The genus *Devosia* was enriched in the D36E and D36EFLC treatments (Supplementary Fig. 3F). These results indicate that inoculation with DC3000 or the derivatives altered the interactions of the rhizosphere microbiome.

## Enrichment of *Devosia* in SL-D36E and SL-D36EFLC is responsible for growth promotion and disease suppression

To identify the key bacteria responsible for the growth promotion and disease suppression functions of SL-D36E and D36EFLC, we

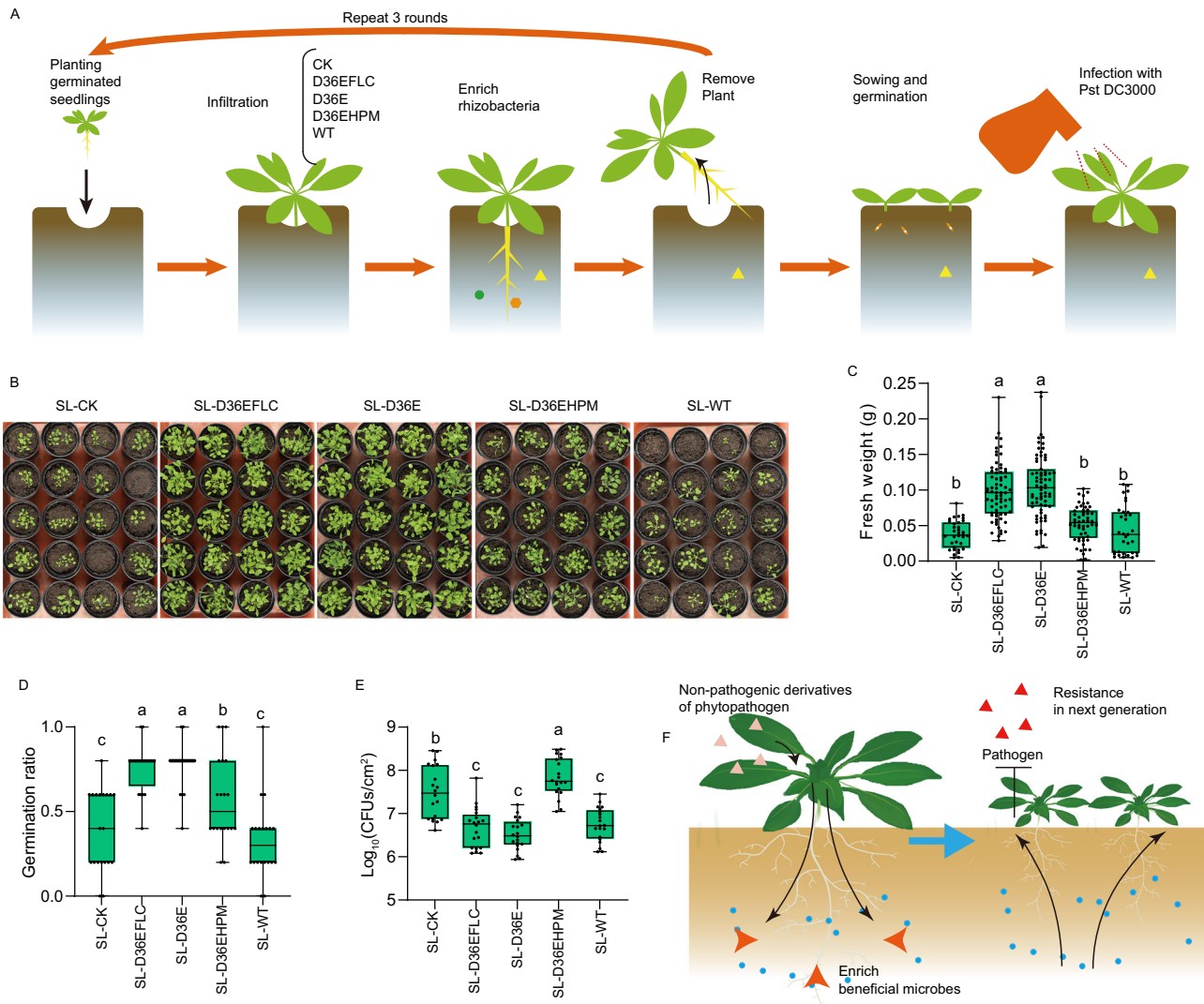

**Fig. 3 | Soil containing the legacy from nonpathogenic strain-infected Arabidopsis is disease suppressive and growth promoting. A** Experimental procedure. **B** Image of plants grown with soil-borne legacy after three rounds of treatment. Each pot was sown with 5 seeds. The images were taken at 5 weeks after sowing. Throughout the experiment, each pot was randomly placed. **C** Fresh weights of plants grown in pre-conditioned soils. The shoot tissues were cut and weighed at 5 weeks post-sowing. The different letters indicate significant differences according to two-sided ANOVA based on Duncan's multiple range test ($\alpha = 0.05$, $n = 34$, 70, 71, 53, 33 independent plant seedlings, $P < 0.001$). **D** Germination ratios in pre-conditioned soils recorded at 5 weeks post-sowing. The different letters indicate significant differences according to two-sided ANOVA based on Duncan's multiple

range test ($\alpha = 0.05$, $n = 20$ independent pots, $P < 0.001$). **E** Colonization of wild-type DC3000 on leaves of 5-week-old Arabidopsis grown in pre-conditioned soils. Colonization was measured by plate counting at 7 days post-inoculation. The different letters indicate significant differences according to two-sided ANOVA based on Duncan's multiple range test ($\alpha = 0.05$, $n = 20$ biologically independent infection experiments, $P < 0.001$). **F** Model of the "cry for help" response elicited by non-pathogenic DC3000 derivatives. The nonpathogenic strain derived from phytopathogens (D36E and D36EFLC) induces a disease-suppressive microbiome in the soil-borne legacy to protect next-generation plants from disease. For boxplots in this figure, median, 25%/75% percentiles, and the highest and lowest values are shown.

further searched for OTUs that were enriched in the growth-promoting, disease-suppressive rhizosphere microbiome (D36E and D36EFLC) (Fig. 4B). Based on a fold-change and q value (FDR) cutoff of >2 and <0.05, respectively, four OTUs were identified (Fig. 4C, Supplementary Fig. 3G–I), among which 2 OTUs (OTU168 and OTU7191) were affiliated with Proteobacteria (Supplementary Table 2). The abundance of OTU7191 in the rhizosphere was significantly (two-sided ANOVA based on Kruskal–Wallis test, $\alpha = 0.05$, $n = 6$ biologically independent samples, $P < 0.001$) increased in both SL-D36E and SL-D36EFLC in comparison with other soils, while that of OTU168 (two-sided ANOVA based on Kruskal-Wallis test, $\alpha = 0.05$, $n = 6$ independent samples, $P < 0.001$) was lower in SL-D36E and SL-D36EFLC (Fig. 4C and Supplementary Fig. 3G). Based on the partial 16 S rDNA sequence, OTU7191 was taxonomically classified into

*Devosia* (Supplementary Table 2). *Devosia* was one of the major groups that was observed in all samples. OTU7191 accounted for the major group of the genus *Devosia* (Fig. 4D), which was also increased in both SL-D36E and SL-D36EFLC (Fig. 4D). Considering the reports that *Devosia* is generally a group consisting of plant-beneficial bacteria[30–33], we hypothesize that OTU7191 from the genus *Devosia* may play a critical role in growth promotion and disease suppression. Therefore, the culturable bacterial strains were isolated from the rhizosphere samples, which were collected from the SL-D36E and SL-D36EFLC treatments, and two isolates, CJK-A8-3 (Genbank: PP325837) and XL339 (Genbank: PP325838), were matched with OTU7191 according to their 16 S rRNA genes (Fig. 4E).

We then cultured these two bacteria in medium and applied a 1:1 mixture of the cell suspension as a consortium to SL-CK and untreated

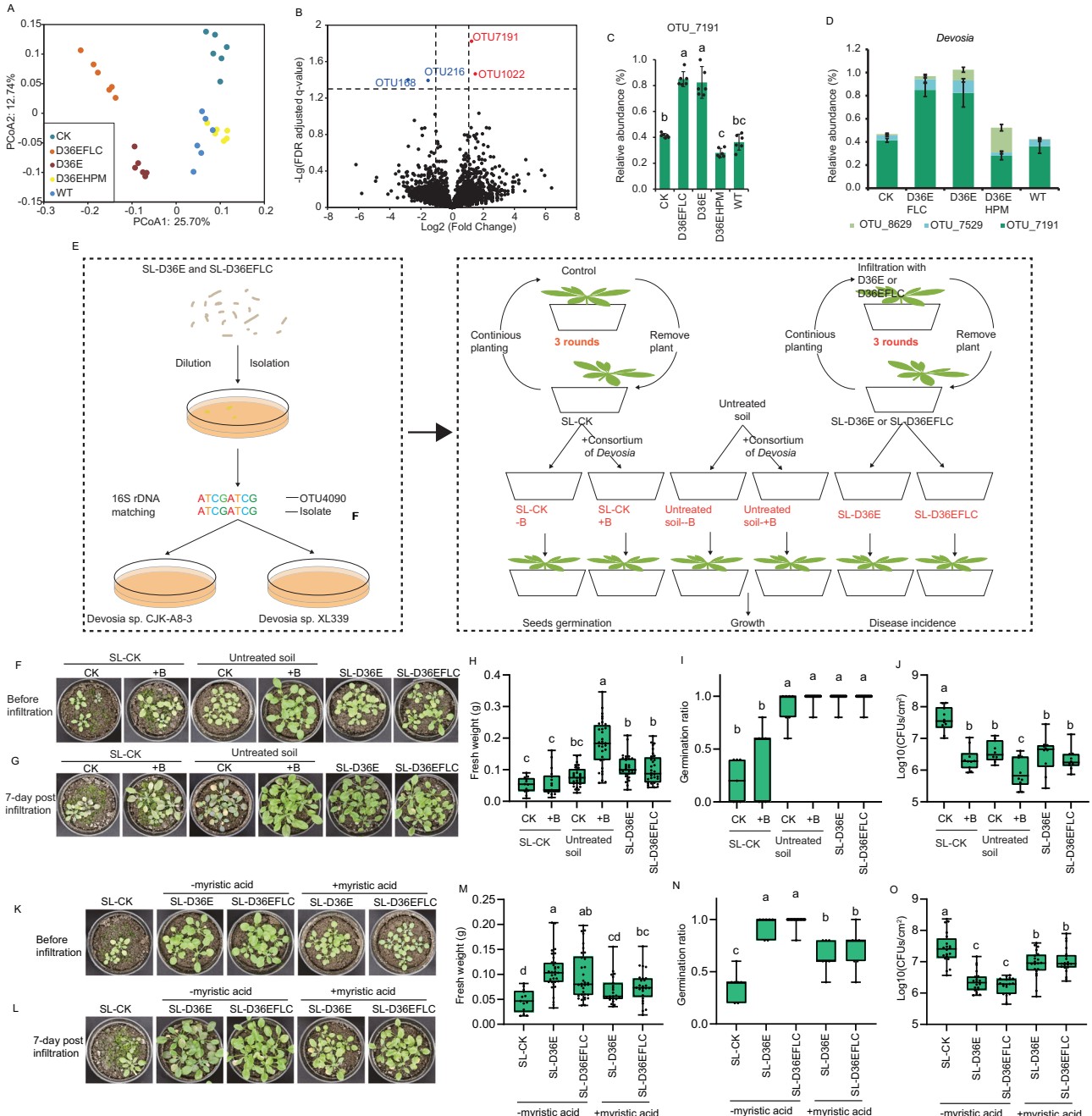

**Fig. 4 | Analysis of the identified OTUs. A** PCoA of the bacterial community at the OTU level via the Bray–Curtis algorithm. **B** Four OTUs that were significantly altered (fold-change > 2, q < 0.05) in the SL-D36E and SL-D36EFLC groups were identified. **C** Relative abundance of OTU_7191. Two-sided ANOVA based on Duncan's multiple range test was used ($\alpha$ = 0.05, $n$ = 6 biologically independent samples, $P$ < 0.001). **D** The *Devosia* composition. $n$ = 6 biologically independent samples. For **C**, **D** Data are presented as the mean values +/- SEMs. (**E**) Workflow of bacterial isolation and functional verification. **F**, **G** Image of plants at 5 weeks after sowing (**F**) and plants infected by DC3000 at 7 days post-infiltration (**G**). "+B" indicates adding a mixture of *Devosia* sp. CK indicates no inoculation. Untreated soil indicates the natural soil without any legacy. **H** Fresh weight of plants measured at 5 weeks post-sowing. Two-sided ANOVA based on Duncan's multiple range test was used ($\alpha$ = 0.05, $n$ = 12, 17, 29, 32, 31, 32 independent plant seedlings, $P$ < 0.001). **I** Germination ratios at 5 weeks post-sowing. Two-sided ANOVA based on Duncan's multiple range test was used ($\alpha$ = 0.05, $n$ = 7 independent pots, $P$ < 0.001). **J** Colonization of DC3000 on leaves of 5-week-old Arabidopsis at 7 days post-inoculation. Two-sided ANOVA based on Duncan's multiple range test was used ($\alpha$ = 0.05, $n$ = 20 biologically independent infection experiments, $P$ < 0.001). **K**, **L** Image of plants taken at 5 weeks post-sowing (**K**) and at 7 days post-infiltration by DC3000 (**L**). **M** Fresh weight of plants measured at 5 weeks post-sowing. Two-sided ANOVA based on Duncan's multiple range test was used ($\alpha$ = 0.05, $n$ = 13, 33, 34, 23, 25 independent plant seedlings, $P$ < 0.001). **N** Germination ratios at 5 weeks post-sowing. Two-sided ANOVA based on Duncan's multiple range test was used ($\alpha$ = 0.05, $n$ = 7 independent pots, $P$ < 0.001). **O** Colonization of DC3000 on leaves of 5-week-old Arabidopsis at 7 days post-inoculation. Two-sided ANOVA based on Duncan's multiple range test was used ($\alpha$ = 0.05, $n$ = 20 biologically independent infection experiments, $P$ < 0.001). For boxplots in this figure, median, 25%/75% percentiles, and the highest and lowest values are shown. For **C**, **H–J** and **M–O**, the different letters indicate significant differences.

soil (Fig. 4E) and evaluated seed germination, fresh weight and disease resistance (Fig. 4F–J). Indeed, the application of the consortium of *Devosia* sp. CJK-A8-3 and *Devosia* sp. XL339 to the untreated soil strongly increased (two-sided ANOVA, $\alpha = 0.05$, $n = 12, 17, 29, 32, 31, 32$ independent plant seedlings, $P < 0.001$) plant fresh weight (Fig. 4F and H). The ratio of seed germination in SL-CK applied with the consortium was lower (two-sided ANOVA, $\alpha = 0.05$, $n = 7$ independent pots, $P < 0.001$) than those in SL-D36E and SL-D36EFLC (Fig. 4F and I). The application of the consortium to SL-CK could not rescue the low fresh weight, which might be due to other cumulative negative effects of continuous planting. As expected, the application of the consortium to both SL-CK and untreated soil decreased (two-sided ANOVA, $\alpha = 0.05$, $n = 20$ biologically independent infection experiments, $P < 0.001$) the colonization of wild-type DC3000 on plant leaves (Fig. 4G and J). This evidence demonstrated that the consortium of *Devosia* sp. CJK-A8-3 and *Devosia* sp. XL339 could contribute to plant growth promotion and disease suppression effects of the soil and further confirmed that the enrichment of OTU7191, which belongs to *Devosia*, contributes to the growth promotion and disease suppression effects of SL-D36E and SL-D36EFLC.

The abundance of the *Devosia* genus in the soil and rhizosphere has been reported to be correlated with root exudate compounds[33], and we wondered which root exudate compound was responsible for enriching this group of bacteria after D36E and D36EFLC inoculation. We correlated the abundance of OTU7191 with compound abundance in root exudates and found 3 compounds in root exudates that were significantly correlated ($q < 0.01$) with the abundance of the OTU (Supplementary Data 2). The abundance of OTU7191 was negatively correlated with L-malic acid (Spearman, $R = -0.526$, q(BH) = 0.003) and myristic acid (Spearman, $R = -0.633$, q(BH) < 0.001) in root exudates but positively correlated (Spearman, $R = 0.776$, q(BH) < 0.001) with 4-hydroxypyridine (Supplementary Fig. 4A–C). Pure L-malic acid, 4-hydroxypyridine and myristic acid were tested with the isolated *Devosia* strains for their potential effects on biofilm formation, the growth curve and motility, which are important features for colonization in the rhizosphere. *Devosia* sp. CJK-A8-3 could not form structured biofilms in MSgg medium, which is a common medium used for biofilm formation (Supplementary Fig. 4D), whereas *Devosia* sp. XL339 formed a structured biofilm in MSgg medium (Supplementary Fig. 4E). Importantly, biofilm formation by *Devosia* sp. XL339 was inhibited by myristic acid in a dose-dependent manner (Supplementary Fig. 4E), and a growth curve assay demonstrated that myristic acid could significantly suppress the growth of *Devosia* sp. CJK-A8-3 (Supplementary Fig. 4F–K). In addition, we did not detect significant motility in these two strains (Supplementary Fig. 5). These results implied that the change in the abundance of myristic acid contributed to the enrichment of *Devosia* in SL-D36E and SL-D36EFLC. Hence, we speculated that adding myristic acid to the soil would offset the beneficial effects of the D36E- and D36EFLC-conditioned soil legacy. We then treated SL-D36E and SL-D36EFLC with myristic acid for 1 week and used the resulting soil for planting. We found that plant fresh weights (two-sided ANOVA, $\alpha = 0.05$, $n = 13, 33, 34, 23, 25$ independent plant seedlings, $P < 0.001$) and germination ratios (two-sided ANOVA, $\alpha = 0.05$, $n = 7$ independent pots, $P < 0.001$) were lower in myristic acid-treated than in the untreated SL-D36E and SL-D36EFLC (Fig. 4K–N), while leaf colonization (two-sided ANOVA, $\alpha = 0.05$, $n = 20$ biologically independent infection experiments, $P < 0.001$) by DC3000 was higher (Fig. 4O), indicating that adding myristic acid to SL-D36E and SL-D36EFLC impaired their beneficial effect. These results demonstrate that a decrease in myristic acid in root exudates contribute to an increase in *Devosia* and enriches the beneficial soil-borne legacy during the D36E- and D36EFLC-triggered "cry for help" response.

## The D36E and D36EFLC metabolite-triggered plant "cry for help" response is quenched upon ETI

We then speculated that some secreted molecules in the supernatant of the D36E or D36EFLC culture might be responsible for inducing the plant "cry for help" response and that D36EHPM may block this response, given that SL-D36EHPM did not show a beneficial effect. We cultured D36E and D36EFLC with plant extracts, collected the supernatants (hereafter D36E/S and D36EFLC/S) and treated plant leaves with the supernatants. In addition, we inoculated leaves with living D36EHPM and performed supernatant treatment. These treatments were applied for three generations to enrich the soil-borne legacy. The rhizosphere microbiome composition of legacy-containing soil was then analyzed by amplicon sequencing. PCoA revealed compositional dissimilarities between soil containing the soil-borne legacy from D36EFLC supernatant- and D36E supernatant-treated plants (hereafter referred to as SL-D36EFLC/S and SL-D36E/S) and SL-CK (Fig. 5A). However, the microbiome of soil containing the soil-borne legacy of plants treated with D36EHPM and D36E/S clustered with the microbiome of soil containing the soil-borne legacy from plants treated with D36EHPM and D36EFLC/S in PCoA (Fig. 5A), indicating that D36EHPM treatment may block the effect of D36E and D36EFLC on the rhizosphere microbiome. Importantly, significant enrichment (two-sided ANOVA based on the Kruskal–Wallis test, $\alpha = 0.05$, $n = 6$ biologically independent samples, $P < 0.001$) of *Devosia* was observed in SL-D36E/S and SL-D36EFLC/S but not under the treatment with D36EHPM coinoculation (Fig. 5B).

We then evaluated plant growth, seed germination and disease incidence. As expected, Arabidopsis in SL-D36EFLC/S and SL-D36E/S showed increases in fresh weight (two-sided ANOVA, $\alpha = 0.05$, $n = 33, 72, 72, 45, 40, 40, 72$ independent plant seedlings, $P < 0.001$), the germination ratio (two-sided ANOVA, $\alpha = 0.05$, $n = 20$ independent pots, $P < 0.001$) and disease resistance (two-sided ANOVA, $\alpha = 0.05$, $n = 20$ biologically independent infection experiments, $P < 0.001$), as did the plants in SL-D36EFLC (Fig. 5C–G). However, the treatment of D36EFLC/S and D36E/S with live D36EHPM did not induce a beneficial effect of the soil-borne legacy (Fig. 5C–G). These results demonstrated that using the metabolites of D36E and D36EFLC in the absence of live bacterial cells is sufficient to induce the plant "cry for help" response, and live D36EHPM could quench the induction. Moreover, because D36E and D36EFLC are Arabidopsis PTI inducing strains, and D36EHPM quenched the "cry for help", we wandered whether PTI trigger while ETI repress the "cry for help" response. We then treated leaves with flg22 or heat-killed wild-type DC3000, or inoculated leaves with D36EFLC/S and alive D36EavrRpt2 together, a D36E derivative with introduced *avrRpt2* that encodes a well-known effector that triggered ETI in Arabidopsis, these treatments were applied for three generations to enrich the soil-borne legacy. Indeed, Arabidopsis grown in soil containing the soil-borne legacy from flg22- and heat-killed WT DC3000-treated plants (hereafter referred to as SL-flg22 and SL-heat-killed) both showed significant increases in fresh weight (two-sided ANOVA, $\alpha = 0.05$, $n = 36, 52, 52, 62, 35, 37$ independent plant seedlings, $P < 0.001$), the germination ratio (two-sided ANOVA, $\alpha = 0.05$, $n = 20$ independent pots, $P < 0.001$) and disease resistance (two-sided ANOVA, $\alpha = 0.05$, $n = 20$ biologically independent infection experiments, $P < 0.001$), in a less extent as in SL-D36EFLC/S (Fig. 5H–L), indicating that establishment of PTI is sufficient to trigger the "cry for help" response. In addition, neither the treatment of D36EFLC/S with live D36EavrRpt2 or live D36EavrRpt2 alone induced a beneficial effect of the soil-borne legacy (Fig. 5H–L). Both the D36EavrRpt2 and the D36EHPM quenched the D36EFLC/S induced beneficial effect of the soil borne legacy, suggesting that ETI can block the "cry for help" response triggered by PTI.

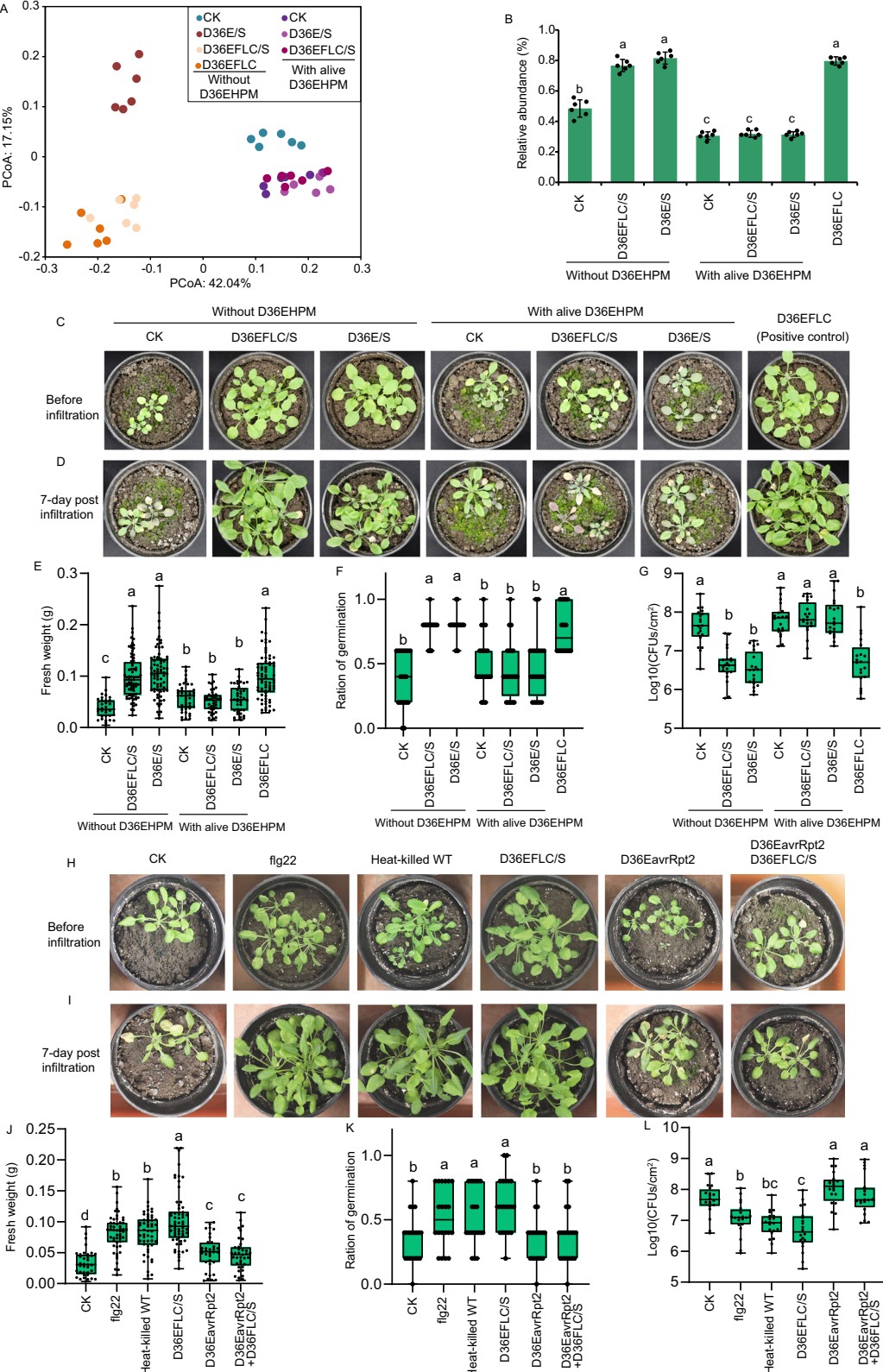

## Discussion

Recruiting a beneficial rhizosphere microbiome by plants through the "cry for help" mechanism is critical for plants to defend themselves from pathogens[1] and thus to form the long-lasting disease suppressive property of the soil[9]. However, the pathogenicity of the elicitor triggering the "cry for help" response has limited the application of the elicitor in agriculture. Here, we used a series of *Pst* DC3000 derivatives

and wild-type DC3000 to investigate the response of the rhizosphere microbiome. We revealed that the soil legacy of D36E- and D36EFLC-treated Arabidopsis was as disease-suppressive as that of wild-type DC3000-treated Arabidopsis. We then identified the key microbial group and the corresponding recruitment signal in root exudates responsible for the disease-suppressive and growth-promoting activity of the soil legacy. We also revealed that the metabolites of D36E and

**Fig. 5 | Inhibitory effect of live D36EHPM on the "cry for help" response elicited by the metabolites of D36E and D36EFLC. A** PCoA of the bacterial community at the OTU level via the Bray–Curtis algorithm. **B** Relative abundance of *Devosia* in the rhizosphere microbiome. Two-sided ANOVA based on the Kruskal–Wallis test was used ($\alpha = 0.05$, $n = 6$ biologically independent samples, $P < 0.001$). Data are presented as the mean values +/- SEMs. **C**, **D** Image of plants grown in legacy-containing soil taken at 5 weeks post-sowing (**C**) and at 7 days post-infiltration by wild-type DC3000 (**D**). **E** Fresh weight of plants grown in soils measured at 5 weeks post-sowing. Two-sided ANOVA based on Duncan's multiple range test was used ($\alpha = 0.05$, $n = 33, 72, 72, 45, 40, 40, 72$ independent plant seedlings, $P < 0.001$). **F** Germination ratios in soils recorded at 5 weeks post-sowing. Two-sided ANOVA based on Duncan's multiple range test was used ($\alpha = 0.05$, $n = 20$ independent pots, $P < 0.001$). **G** Colonization of wild-type DC3000 on leaves of 5-week-old Arabidopsis grown in soils. Colonization was measured by plate counting at 7 days post-inoculation. Two-sided ANOVA based on Duncan's multiple range test was used

($\alpha = 0.05$, $n = 20$ biologically independent infection experiments, $P < 0.001$). **H**, **I** Image of plants grown in legacy-containing soil. The images were taken at 5 weeks post-sowing (**H**) and at 7 days post-infiltration by wild-type DC3000 (**I**). **J** Fresh weight of plants grown in soils measured at 5 weeks post-sowing. Two-sided ANOVA based on Duncan's multiple range test was used ($\alpha = 0.05$, $n = 36, 52, 52, 62, 35, 37$ independent plant seedlings, $P < 0.001$). **K** Germination ratios in soils recorded at 5 weeks post-sowing. Two-sided ANOVA based on Duncan's multiple range test was used ($\alpha = 0.05$, $n = 20$ independent pots, $P < 0.001$). **L** Colonization of wild-type DC3000 on leaves of 5-week-old Arabidopsis at 7 days post-inoculation. Two-sided ANOVA based on Duncan's multiple range test was used ($\alpha = 0.05$, $n = 20$ biologically independent infection experiments, $P < 0.001$). For boxplots in this figure, median, 25%/75% percentiles, and the highest and lowest values are shown. For **B**, **E**–**G** and **J**–**L**, the different letters indicate significant differences.

D36EFLC and flg22 alone are sufficient to induce a "cry for help" response, which can be quenched by inoculation of either D36EHPM or D36EavrRpt2. This study demonstrates the ability of nonpathogenic strains and their MAMPs to act as elicitors to induce the formation of a disease-suppressive soil legacy (Fig. 3F), which can potentially support agricultural applications.

Generating a disease-suppressive soil-borne microbiome would be an environmentally safe and logically feasible solution for the lasting control of plant disease[9,34]. The plant "cry for help" theory, indicating that plants adapt to biotic stress by changing their root exudation chemistry to assemble health-promoting rhizosphere microbiomes, explains one of the mechanisms by which the disease-suppressive microbiome is formed[7,35]. For instance, it was previously shown that pretreating soil with downy mildew-infected plants led to reduced susceptibility to the disease in a subsequent generation of plants grown in that soil[4]. However, the elicitors and the detailed signaling pathway responsible for rhizosphere microbe recruitment during the "cry for help" process remain to be clarified. This limitation has restricted its application in agriculture given that the "cry for help" response has mostly been observed in plant interactions with plant pathogens, which act as elicitors. Here, we found that treating plants with metabolites of nonpathogenic *Pst* DC3000 derivatives (D36E or D36EFLC) that retained only the potential harmless elicitor or flg22 alone induced the host to form a soil-borne legacy with a long-lasting disease-suppressive function (Fig. 3F). In this study, we demonstrated that the recruitment of beneficial microbes, which was previously considered to result from pathogen infestation[4,36], is elicited by pathogen-derived signals that are not necessary for disease occurrence. This finding has special significance because it supports the possibility of developing a plant-friendly elicitor to trigger the "cry for help" response and to be used in agricultural production.

Plants can dramatically modulate the composition of root exudates upon pathogen infestation[6,11,37], which generally functions against the pathogen by regulating the rhizosphere microbiome[7,38]. The success of defense against infection by foliar pathogens is determined by the establishment of plant PTI and ETI. However, how the immune response in leaves affects root exudate-microbiome interactions is not clear. Systemic immune responses to aboveground pests and pathogens have been reported to alter root interactions with belowground microbes in an SA-dependent manner[39,40]. A previous study also showed that Arabidopsis plants grown in DC3000-infected Arabidopsis-conditioned soil accumulate jasmonic acid[5], suggesting the JA-signaling may also play a role in affecting root exudate-microbiome, which nonetheless need further investigations. Accessions of *Arabidopsis thaliana* show variation in defense phytohormone profiles after infection, but they share similar root-associated microbiota[41]. Lebeis et al. showed that the foliar defense phytohormone salicylic acid modulates the colonization of the root microbiome, and they

concluded that plant immune signaling drives selection of the microbiome[40]. In this study, we found that a deficiency of all effectors and major MAMPs did not compromise the major root exudate-modulating activity of DC3000. Nonetheless, metabolites of D36E and D36EFLC were sufficient to affect exudates and rhizosphere microbiomes. It worth nothing that some weak MAMPs, such as T3SS pilus[42], were still retained in the supernatants of D36E and D36EFLC. In conclusion, these results suggest that establishment of PTI in leaves can trigger the "cry for help", but it does not exclude the possibility that some non-MAMPs metabolites might also contribute to trigger the "cry for help" response. Interestingly, the inoculation of D36EHPM or D36EavrRpt2 blocked the induction of the plant "cry for help" response by D36EFLC/S, and we suggest that the occurrence of ETI may suppress the "cry for help" response. We propose that the occurrence of ETI can serve as a signal of success in combating pathogens. These results strongly suggest that further investigations will identify a harmless molecule in the supernatant of D36E and D36EFLC that is able to induce plant "cry for help" responses. However, our results cannot exclude the possibility that those ETI-eliciting derivatives may cause a plant stunting which in turn affect photosynthetic carbon production and root exudation, and therefore repress the recruitment of beneficial microbes.

In conclusion, we revealed the structure of the microbiome legacy from plants treated with metabolites of nonpathogenic DC3000 derivatives and verified its disease-suppressive and growth-promoting properties. We anticipate this strategy to be a new direction for partially overcoming continuous cropping problems.

## Methods
### Bacterial strains
*Pst* DC3000 and its derivatives used in this study (Supplementary Table 3) were cultured as described previously[23]. Generally, the WT and derivatives were grown in King's B (KB) agar media at 28 °C overnight; when necessary, spectinomycin (Spc), kanamycin (Kan) and rifampicin at a final concentration of 50 mg/L were added to the media. The bacterial cells were subsequently harvested from the Petri dishes and diluted in 10 mM $MgCl_2$ for further use.

### Plants
*Arabidopsis thaliana* ecotype Columbia (Col-0) was used in this study. Col-0 seeds were surface-sterilized with 75% (vol/vol) ethanol followed by 2% (vol/vol) NaClO, after which they were placed in square Petri dishes containing 1/2-strength Murashige and Skoog (MS) medium supplemented with 2% (wt/vol) sucrose and 0.8% (wt/vol) agar and then stratified for 2 days at 4 °C in the dark. Thereafter, the Petri dishes with the seedlings were positioned vertically and transferred to a growth chamber under a 16-h light and 8-h dark photoperiod at 22 °C. Seven days later, the seedlings were transplanted to 6-well plates containing liquid 1/2-strength MS media supplemented with 0.5%

sucrose or to pots containing a vermiculite:peat soil mixture under a humidity of 60%.

## Plant hypersensitive response assays

To test the plant hypersensitive response, 4-week-old plants grown in vermiculite:peat were inoculated with *Pst* DC3000 or the derivatives at $10^6$ cells/mL as described previously[23]. At 6 days post-inoculation, the plant hypersensitivity response was recorded by directly visualizing the degree of leaf collapse. This experiment was repeated twice, with 9 replicates for each treatment.

## Bacterial infection assays for testing the virulence of the DC3000 derivatives

Syringe-infiltration was performed to evaluate the bacterial infection in Arabidopsis leaves. Briefly, DC3000 and the derivatives were diluted to $10^5$ cells/mL and infiltrated to leaves, respectively. The infiltrated plants were kept under high humidity by covering the plant with a plastic membrane for 1 day. For quantification the bacterial density in leaves, Arabidopsis leaves were surface sterilized in 75% ethanol and washed in sterile water. Leaf disks were taken and ground in 10 mM $MgCl_2$. Bacterial densities were determined at 3 days post-inoculation by dilutions and plating onto King's B (KB) agar media containing 50 mg/L rifampicin.

## ROS measurements

For the measurement of plant ROS, bacterial suspensions of *Pst* DC3000 and the derivatives at $5 \times 10^7$ cells/mL were applied to the leaves of 4-week-old Arabidopsis seedlings. At 15 h post-inoculation, the leaves were excised and immersed in a 2% (wt/vol) nitro blue tetrazolium (NBT) solution for 2 h under light and destained with 70% ethanol at 70 °C[43]. Six replicates were included for each treatment, and the entire ROS experiment was repeated twice.

To quantitively measure ROS in plant leaves in response to wild-type DC3000 and derivatives thereof, bacterial suspensions were infiltrated into the leaves of 4-week-old *Arabidopsis* seedlings at $5 \times 10^7$ CFU/mL, and 10 mM $MgCl_2$ was infiltrated into the leaves as a mock control. Subsequently, leaf discs were excised and analyzed for ROS by using L-012 following the method described by Wei et al. [43]. Six replicates were included for each treatment.

## Plant gene transcription assay

The leaves of 4-week-old Arabidopsis plants grown in MS medium were infiltrated with wild-type DC3000 or derivatives thereof. The leaves in each treatment were flash-frozen in liquid nitrogen at 3 h post-inoculation and then ground, and the RNA was extracted using an RNeasy Plant Mini kit (Qiagen, Hilden, Germany). The extracted RNA was evaluated on a 1% agarose gel, and RNA concentration and quality (A260/A280) were determined by a NanoDrop ND-2000 spectrophotometer (NanoDrop). The RNA was reverse transcribed into cDNA by using a PrimeScript RT Reagent Kit with gDNA Eraser (TaKaRa, Dalian, China). A SYBR® Premix Ex Taq™ Kit (Takara, Dalian, China) was used for qRT–PCR in conjunction with a QuantStudio 6 Flex system (Applied Biosystems, Foster City, CA, USA). The following PCR program was used: denaturing for 30 s at 95 °C, followed by 40 cycles of 5 s at 95 °C and 34 s at 60 °C. The primers used are listed in Supplementary Table 4. The $2^{-\Delta\Delta CT}$ method was used to analyze the real-time PCR data, and the *ACTIN2* gene was included as an internal control. Six independent replicates were included for each treatment.

## Root exudate analysis

Leaves of 4-week-old Arabidopsis plants were inoculated with bacterial suspensions of D36EFLC, D36E, D36EHPM and WT at $5 \times 10^7$ cells/mL, respectively. Subsequently, three plants were transplanted to each well of sterile six-well plates containing water–agar media (1% agar) and allowed to grow in a growth chamber under a 16-h light and 8-h dark

photoperiod at 22 °C for root exudate collection. After 3 days, the plants were removed, and the water–agar media in all the wells of one plate, including the root-attached agar, were collected as one replicate. Six replicates were included for each treatment. The agar containing the root exudates was checked for contamination and freeze dried to obtain powders, which were then subjected to gas chromatography–time-of-flight–mass spectrometry (GC–TOF–MS) and analyzed as described previously[11]. Briefly, the root exudate powder was placed in a 1.5 mL Eppendorf (EP) tube, 20 μL of methoxyamination hydrochloride (dissolved in pyridine at a final concentration of 20 mg/mL) was added to the tube, and the mixture was incubated for 30 min at 80 °C. Then, 30 μL of the BSTFA + TMCS reagent (N,O-bis(trimethylsilyl) trifluoroacetamide with 1% trimethylchlorosilane, v/v) was added to each sample aliquot, and each mixture was incubated for 1.5 h at 70 °C. Five microliters of saturated fatty acid methyl esters (FAMEs) dissolved in chloroform was added to the sample after it was allowed to cool to room temperature. All the samples were analyzed with a gas chromatograph system coupled to a Pegasus HT time-of-flight mass spectrometer. GC–TOF–MS analysis was performed using an Agilent 7890 gas chromatograph system coupled to a Pegasus HT time-of-flight mass spectrometer. The system utilizes a DB-5MS capillary column coated with 5% diphenyl cross-linked with 95% dimethylpolysiloxane (30 m×250 μm inner diameter, 0.25 μm film thickness; J&W Scientific, Folsom, CA, USA). A 1 μL aliquot of the sample was injected in splitless mode. Helium was used as the carrier gas, the front inlet purge flow was 3 mL/min, and the gas flow rate through the column was 1 mL/min. The initial temperature was maintained at 50 °C for 1 min and then raised to 310 °C at a rate of 10 °C min for 8 min. The injection, transfer line, and ion source temperatures were 280, 280, and 250 °C, respectively. The energy was -70 eV in electron impact mode. Mass spectrometry data were acquired in full-scan mode with an m/z range of 50-500 at a rate of 12.5 spectra/second after a solvent delay of 6.17 min. Chroma TOF 4.3X software (LECO Corporation) and the LECO-Fiehn Rtx5 database were used for extracting the raw peaks, filtering the data baselines, calibrating the baseline, aligning the peaks, analyzing the deconvolution, identifying the peaks and integrating the peak areas[44]. Both mass spectrum matching and retention index matching were considered in metabolite identification. No internal standards were used. Five equally pooled samples from all biological samples were used as QC samples during measurement, peaks detected in <50% of the quality control (QC)-filtered samples or QC-filtered samples with a relative standard deviation (RSD) > 30% were removed[45]. The order for sample preparation and data collection of biological samples was randomized. The root exudate composition profile obtained with peak height was normalized by the sum of all the compounds in each sample. The data were then normalized by the mean-centered method. Principal component analysis was performed via R (version 3.6), and nonmetric multidimensional scaling (NMDS) analysis was performed with the phyloseq package of R (version 3.6) based on pairwise Bray–Curtis similarity.

## Soil property analysis

The soil used in this study was collected from the Hulunber Grassland Ecosystem Observation and Research Station located at the Xiertala farm, Inner Mongolia, China (49.414° N, 120.078° E). The soils were air dried, sieved and homogenized using standard methods. Soil pH and conductivity were measured by the electrode method at a soil:water ratio of 1:5. Soil organic matter and organic carbon contents were determined via the potassium dichromate volumetric method and the heating method, respectively[46]. Available phosphorus was extracted with sodium bicarbonate, total phosphorus was resolved by NaOH solution, and both were subsequently measured by the molybdenum blue method. Available potassium was extracted with ammonium acetate and measured via inductively coupled plasma-atomic emission

spectroscopy (ICP–AES). Total potassium was resolved with an NaOH solution and measured via flame photometry. Total nitrogen was determined by Kjeldahl digestion[47], and soil cation exchange was measured via NaAc-ICP–AES assays.

## Assays of plants grown in legacy soils

To generate soil containing the soil-borne legacy, five seven-day-old Arabidopsis seedlings were transplanted into each pot containing 120 g of soil and allowed to grow for 20 days. Then, plant leaves were dipped into suspensions of *Pst* DC3000 or the derivatives ($5 \times 10^7$ cells/mL) or a solution of 10 mM $MgCl_2$ (control solution), and the extra cell suspensions on the leaves were dried to avoid contamination of the soil. All pots were randomly placed and covered to maintain 100% humidity. Plant disease or immune responses were allowed to develop for 7 days, after which the plants were removed. New 7-day-old seedlings were transplanted into pots filled with soil containing the legacy for the next generation. The plants and soils were then repeatedly treated in the same way as those of the previous generation. After 3 generations, soil containing the soil-borne legacies from D36EFLC-, D36E-, D36EHPM-, WT-infected or uninfected Arabidopsis (hereafter referred to as SL-D36EFLC, SL-D36E, SL-D36EHPM, SL-WT and SL-CK, respectively) was generated.

After then, Arabidopsis seeds were sown in pot filled with soil containing the legacy, the seedlings were allowed to grow for 5 weeks. For evaluating the germination ratio, each treatment includes 20 pots with five seedlings in each. The germination ratio in each pot was recorded at 10 days post-sowing, and the average germination ratio of the 20 pots for each treatment was calculated. The plant shoots were cut and weighed at 5 weeks post-sowing. For evaluating the disease resistance against DC3000, 20 leaves of 5-week-old plants grown in each soil were infiltrated with a suspension of wild-type *Pst* DC3000. Seven days after inoculation with wild-type D3000, 20 infiltrated leaves were cut into discs and ground, and the ground material was spread on Petri dishes containing agar medium to count CFUs.

To verify the function of the isolated *Devosia* bacteria, soil containing the D36EFLC- and D36E-infected Arabidopsis soil-borne legacy (SL-D36EFLC, SL-D36E) or uninfected Arabidopsis soil-borne legacy (SL-CK) was used to grow Arabidopsis, and the original untreated soil was also used as a control. Five Arabidopsis seeds were sown in each pot. A mixture of equal amounts (bacterial concentrations) *Devosia* sp. CJK-A8-3 and *Devosia* sp. XL339 suspensions was added to SL-CK and untreated soil at the same time as sowing was performed. The final concentration of the bacteria was $10^7$ cells/mL. For evaluating germination ratio and shoot fresh weigh, each treatment includes 7 pots with 5 seeds sown to each pot. The seedlings were allowed to grow for 5 weeks. The germination ratio in each pot was recorded at 10 days post-sowing, and the average germination ratio of the 7 pots was calculated. The plant shoots of all seedlings from 7 pots were cut and weighed at 5 weeks post-sowing. For evaluating the disease resistance against DC3000, 20 leaves of 5-week-old plants grown in 5 pots for each treatment were infiltrated with a suspension of wild-type *Pst* DC3000. Seven days after inoculation, 20 infiltrated leaves were cut into discs and ground, and the ground material was spread on Petri dishes containing agar medium to count CFUs.

To verify the function of myristic acid in the effect of the soil containing the legacy, soil containing the D36EFLC- and D36E-infected Arabidopsis soil-borne legacy (SL-D36EFLC, SL-D36E) or uninfected Arabidopsis soil-borne legacy (SL-CK) was used to grow Arabidopsis. Both SL-D36EFLC and SL-D36E were treated with 40 μM myristic acid via saturated irrigation. For evaluating germination ratio and shoot fresh weigh, each treatment includes 7 pots with 5 seeds sown to each pot. The seedlings were allowed to grow for 5 weeks. The germination ratio in each pot was recorded at 10 days post-sowing, and the average germination ratio of the 7 pots for each treatment was calculated. The

plant shoots of all seedlings from 7 pots were cut and weighed at 5 weeks post-sowing. For evaluating the disease resistance against DC3000, 20 leaves of the 5-week-old plants grown in 5 pots for each treatment were infiltrated with a suspension of wild-type *Pst* DC3000. Seven days after inoculation, 20 infiltrated leaves were cut into discs and ground, and the ground material was spread on Petri dishes containing agar medium to count CFUs.

To study the effect of the metabolites of D36E and D36EFLC on the elicitation of the plant "cry for help" response, D36E and D36EFLC were cultured in liquid KB medium containing plant leaf extracts to collect the supernatant, respectively. Seven-day-old Arabidopsis seedlings were transplanted into pots containing 120 g of soil and allowed to grow for 20 days. Then, plant leaves were dipped into a suspension ($5 \times 10^7$ cells/mL) of D36EFLC, a suspension of D36EHPM ($5 \times 10^7$ cells/mL), the supernatant of D36E (D36E/S), the supernatant of D36EFLC (D36EFLC/S), a suspension of D36EHPM in the supernatant of D36E (D36EHPM + D36E/S), a suspension of D36EHPM in the supernatant of D36E (D36EHPM + D36EFLC/S), or 10 mM $MgCl_2$ (CK), and the excess cell suspension or supernatant remaining on the leaves was dried to avoid contamination of the soil. All pots were randomly placed and covered to maintain 100% humidity. Plants were allowed to grow for 7 days, after which the plants were removed. New 7-day-old seedlings were transplanted into pots filled with soil containing the legacy for the next generation. The plants and soils were then repeatedly treated in the same way as those of the previous generation. After 3 generations, soil containing the legacy from D36EFLC-, D36EHPM-, D36E/S-, D36EFLC/S-, D36EHPM + D36E/S- or D36EHPM + D36EFLC/S-treated Arabidopsis (hereafter referred to as SL-D36EFLC, SL-D36EHPM, SL-D36E/S, SL-D36EFLC/S, SL-D36EHPM + D36E/S, SL-D36EHPM + D36EFLC/S and SL-CK, respectively) was generated. For evaluating germination ratio and shoot fresh weigh, five Arabidopsis seeds were sown into each pot filled with soil containing the legacy and 20 pots were included for each treatment, the seedlings were allowed to grow for 5 weeks. The germination ratio in each pot was recorded at 10 days post-sowing, and the average germination ratio of the 20 pots was calculated. The plant shoots of all seedlings from 20 pots were cut and weighed at 5 weeks post-sowing. For evaluating the disease resistance against DC3000, 20 leaves of the 5-week-old plants grown in 5 pots for each treatment were infiltrated with a suspension of wild-type *Pst* DC3000. Seven days after inoculation, 20 infiltrated leaves were cut into discs and ground, and the ground material was spread on Petri dishes containing agar medium to count CFUs.

To verify the effect of flg22, heat-killed WT DC3000 and D36EavrRpt2 on the elicitation of the plant "cry for help" response, 7-day-old Arabidopsis seedlings were transplanted into pots containing 120 g of soil and allowed to grow for 20 days. Then, plant leaves were infiltrated with flg22, or dipped into a suspension ($5 \times 10^7$ cells/mL) of heat-killed WT DC3000, the supernatant of D36EFLC (D36EFLC/S), a suspension of D36EavrRpt2 (D36EavrRpt2), a suspension of D36EavrRpt2 in the supernatant of D36EFLC (D36EavrRpt2 + D36EFLC/S), or 10 mM $MgCl_2$ (CK), and the excess cell suspension or supernatant remaining on the leaves was dried to avoid contamination of the soil. All pots were randomly placed and covered to maintain 100% humidity. Plants were allowed to grow for 7 days, after which the plants were removed. New 7-day-old seedlings were transplanted into pots filled with soil containing the legacy for the next generation. The plants and soils were then repeatedly treated in the same way as those of the previous generation. After 3 generations, soil containing the legacy was generated. For evaluating germination ratio and shoot fresh weigh, five Arabidopsis seeds were sown into each pot filled with soil containing the legacy and 20 pots were included for each treatment, the seedlings were allowed to grow for 5 weeks. The germination ratio in each pot was recorded at 10 days post-sowing, and the average germination ratio of the 20 pots was calculated. The plant shoots of all

seedlings from 20 pots were cut and weighed at 5 weeks post-sowing. For evaluating the disease resistance against DC3000, 20 leaves of the 5-week-old plants grown in 5 pots for each treatment were infiltrated with a suspension of wild-type *Pst* DC3000. Seven days after inoculation, 20 infiltrated leaves were cut into discs and ground, and the ground material was spread on Petri dishes containing agar medium to count CFUs.

## Rhizosphere soil collection

For collecting rhizosphere soil of the Arabidopsis grown in SL-D36EFLC, SL-D36E, SL-D36EHPM, SL-WT and SL-CK, Arabidopsis seeds were sown in pot filled with the conditioned soil, respectively. Each treatment includes 6 pots with 5 seeds in each pot. The 5-weeks old seedlings were inoculated with wild-type DC3000. Seven days after inoculation, the rhizosphere soil samples were collected following the method reported by Bulgarelli et al. [48]. Arabidopsis plants and the soil were carefully harvested from the pots, and large soil aggregates were removed by shaking the roots. The roots of all plants in the same pot were pooled into a 15 mL Falcon tube containing 2.5 mL of sterile Silwet L-77 amended PBS buffer (PBS-S; 130 mM NaCl, 7 mM $Na_2HPO_4$, 3 mM $NaH_2PO_4$ [pH 7.0], 0.02% Silwet L-77) and washed on a shaking platform for 20 min at 180 rpm. Then, the roots were removed and the washing buffer was subjected to centrifugation ($1500 \times g$, 20 min). The resulting pellet was frozen in liquid nitrogen and stored at -80 °C. Rhizosphere soil collected from all plants grown in one pot were pooled as one replicate. Six replicates were included and sampled for each treatment.

For collecting rhizosphere soil of the Arabidopsis grown in SL-D36EFLC, SL-D36EHPM, SL-D36E/S, SL-D36EFLC/S, SL-D36EHPM + D36E/S, SL-D36EHPM + D36EFLC/S and SL-CK, Arabidopsis seeds were sown in pot filled with the conditioned soil, respectively. Each treatment includes 6 pots with 5 seeds in each pot. The seedlings were allowed to grow for 5 weeks and subsequently inoculated with wild-type DC3000. Seven days after inoculation, rhizosphere soil was collected with the same method described above. Rhizosphere soil collected from all plants grown in one pot were pooled as one replicate. Six replicates were included and sampled for each treatment.

## Rhizosphere microbiome analysis

Total DNA was extracted from 500 mg rhizosphere soil samples using a Power Soil DNA Isolation kit (MoBio Laboratories, Inc., Carlsbad, CA, USA) according to the manufacturer's instructions. The purity and quality of the genomic DNA were checked on 1% agarose gels and a NanoDrop spectrophotometer (Thermo Scientific, 111 Wilmington, DE, USA). The V3-V4 hypervariable region of the bacterial 16 S rRNA gene was amplified with the primers 338 F (5′-ACTCCTACGGGAGGCAGCAG-3′) and 806 R (5′-GGACTACNNGGGTATCTAAT-3′). The 8-digit barcodes were added to the forward and reverse primers. PCR was carried out on a Mastercycler Gradient thermal cycler (Eppendorf) using 25 μL reaction solutions consisting of 12.5 μL of KAPA 2 G Robust Hot Start Ready Mix, 1 μl of forward and reverse primers (5 μM), 5 μl of DNA (with a total template quantity of 30 ng), and 5.5 μl of $H_2O$. The cycling program was 95 °C for 5 min; 28 cycles of 95 °C for 45 s, 55 °C for 50 s and 72 °C for 45 s; and a final extension at 72 °C for 10 min. The PCR products were subsequently purified using an Agencourt AMPure XP Kit. Deep sequencing was performed on a MiSeq platform at Allwegene Company (Beijing). The raw data generated have been deposited in the NCBI database under accession code PRJNA803271. The sequences were first separated using sample-specific barcode sequences. Pear (v0.9.6)[49] software was used to splice and remove the sequences with a low-quality score (≤ 20), those containing ambiguous bases and those that did not exactly match the primer sequences or barcode tags. Vsearch (v2.7.1)[50] software was used to remove sequences with lengths <230 bp. Chimeric sequences were removed via the uchime method according to the Gold Database[51]. After removing singletons, the sequences were clustered into

OTUs at a similarity level of 97% using the Uparse algorithm of Vsearch (v2.7.1) software[52]. The Ribosomal Database Project (RDP) Classifier tool was used to classify all the sequences into different taxonomic groups[53]. All OTUs annotated as chloroplasts, mitochondria or archaea were removed. Based on the rarefied OTU table (Supplementary data 3), the richness and diversity indices were calculated. The beta diversity of the bacterial community was analyzed using the PCoA method. OTUs with significantly different abundance between each treatment were identified by ANOVA based on the Kruskal–Wallis test with a false discovery rate (FDR) <0.01.

## Biofilm formation, growth curve and motility assay

The biofilm formation assay was performed as described by Liu et al.[54]. *Devosia* strains were cultured in TSB medium (5 g/L NaCl, 2.5 g/L $K_2HPO_4$, 2.5 g/L glucose, 17 g/L tryptone, 3 g/L phytone, pH 7.3) until the $OD_{600}$ reached 0.6. The bacterial cells were collected by centrifugation, washed with sterile water and suspended in biofilm formation medium (MSgg medium: 5 mM potassium phosphate, 100 mM morpholinepropanesulfonic acid [MOPS], 2 mM $MgCl_2$, 700 mM $CaCl_2$, 50 mM $MnCl_2$, 50 mM $FeCl_3$, 1 mM $ZnCl_2$, 2 mM thiamine, 0.5% glycerol, 0.5% glutamate, 50 mg/ml tryptophan, 50 mg/ml phenylalanine, and 50 mg/ml threonine, pH 7.0) at a final $OD_{600}$ of 1.0. Biofilm experiments were performed in a 48-well microtiter plate with 1 mL of MSgg medium in each well. In each well, 10 μL of bacterial suspension and the tested chemical were added. Each treatment includes 4 replicates.

The motility assay was performed following the method described by Inoue et al.[55]. Briefly, *Devosia* strains were cultured in LB medium until the $OD_{600}$ reached 0.6 and then inoculated into Petri dishes containing semisolid TSB medium (0.6% (w/v) Eiken agar (Eiken, Nogi-machi, Japan)) by using sterilized toothpicks. The Petri dishes were incubated at 37 °C for 4 h to allow the bacteria to swim and then dried to terminate the motility. Subsequently, the Petri dish was incubated overnight to observe the motility circle. Four replicates were included for each treatment.

The growth curves of the *Devosia* strains in LB medium were measured. *Devosia* strain was inoculated to LB medium to an initial concentration of $OD_{600} = 0.01$. Each chemical was added to the medium with a final concentration of 1 μM, 5 μM, 8 μM, 40 μM and 200 μM, respectively. $OD_{600}$ measurement was performed every hour using the Bioscreen C system. Four replicates were included for each treatment.

## Reporting summary

Further information on research design is available in the Nature Portfolio Reporting Summary linked to this article.

## Data availability

The 16 S raw data of rhizosphere microbiome generated have been deposited in the NCBI database under accession code PRJNA803271. The 16 S rRNA sequence of *Devosia* sp. strain CJK-A8-3 and *Devosia* sp. strain XL339 have been deposited in GenBank with the accession code PP325837 and PP325838, respectively. The metabolomics mass spectrometry raw data are only available for request because there are other uses of the mass spectrometry raw data. Source data are provided with this paper.

## Code availability

The code used to analyze the data are provided in the supplementary information.

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

## Acknowledgements

This work was funded by the National Key Research and Development Program (2022YFF1001804, R.Z., 2021YFF1000400, Y.L. and R.Z., 2021YFD1900300, W.X.), the Innovation Program of Chinese Academy of Agricultural Sciences (CAAS-CSAL-202302, Y.L.), the Central Public-interest Scientific Institution Basal Research Fund (No. Y2022QC15, Y.L.) and the Agricultural Science and Technology Innovation Program (CAAS-ZDRW202308, Y.L.). We thank Prof. Hailei Wei of the Institute of Agricultural Resources and Regional Planning, Chinese Academy of Agricultural Sciences for providing us the strains D36E, D36EFLC and D36EHPM, and Prof. Xiu-fang Xin of Institute of Plant Physiology and Ecology, Chinese Academy of Sciences for providing us the strain D36EavrRpt2.

## Author contributions

Conceptualization, Y.L. and R.Z.; Formal Analysis, H.Z., W.X.; Investigation, Y.L., H.Z. J.W., W.G. and X.Sun; Verification, Q.X. and W.X.; Writing -Original Draft, Y.L.; Writing-Review & Editing, Y.L., W.X. and R.Z.; Visualization, X.Shu and Y.M.; Supervision, Y.L. W.X. and R.Z.; Project Administration, Q.S.; Funding Acquisition, Y.L. and R.Z.

## Competing interests

The authors declare no competing interests.
