## [Peer Review File · Nature Communications]

Reviewers' Comments:

Reviewer #1:

Remarks to the Author:

This manuscript reports that infection with avirulent strains of the bacterial pathogen *Pseudomonas syringae* DC3000 leads to the enrichment of beneficial soil microbes that promote growth and disease resistance of *Arabidopsis thaliana*. The authors identified OTUs belonging to the genus *Devosia* that can promote these plant traits. Because of this, they propose that avirulent pathogens can use as plant vaccines.

The authors first characterized *Arabidopsis* immune responses to different strains of *P. syringae* DC3000 including wild-type, type III effector-deficient (D36E), Flc-deficient in addition to D36E (D36EFLC), and D36E carrying an effector HopM1 (D36EHPM). Then, they performed three rounds of leaf treatment with these strains, followed by measurement of root exudates, shoot fresh weight, and resistance against wild-type *P. syringae* DC3000. They observed that in particular, D36E and D36EFLC conditioned soil microbes promoted plant growth and resistance. They identified differentially abundant OTUs and isolated bacterial strains corresponding to these OTUs. Strains belonging to *Devosia* OTUs could recapitulate effects of D36E and D36EFLC-conditioned soil on plant growth and disease resistance promotion.

Pathogen-infection-triggered enrichment of beneficial bacteria that promote disease resistance has been well documented including *P. syringae* (Berg and Koskella *Curr Biol* 2018; Yuan et al *Microbiome* 2018). Thus, the novelty in this manuscript is that this can also happen by avirulent pathogens. Identification of OTUs belonging to *Devosia* for promoting growth and resistance is also a novel finding of this paper.

While this study adds to our understanding of "cry-for-help", it remains unclear how avirulent pathogens do this. To this end, it would be highly informative if the authors perform a series of experiments using immune elicitors such as flg22. This answers questions about whether plant immune activation by avirulent (or virulent) pathogens is responsible for cry-for-help or not. Related to this, it remains unclear whether avirulent strains promote the enrichment of beneficial bacteria directly via bacteria-bacteria interactions or through plant immune activation or both.

In summary, their findings contribute to the research community, but the mechanistic depth should be improved. In addition, there are points that the authors should consider to improve this study.

(Major comments)

1. DC3000, which contains HopM1, does not trigger ETI in *Arabidopsis*. Why does D36EHPM trigger ETI? There is no evidence that HopMI triggers ETI in *Arabidopsis* (the authors need to find the receptor for HopM1 in *Arabidopsis*). Furthermore, D36EFLC and DC3000 also trigger PTI. Thus, it is strange to categorize these strains as in Fig1. Defining threat level by measuring plant immune responses does not make sense. If the authors really want to define it, they can measure fitness affected by infection with different strains.
2. As mentioned above, it remains unclear what is responsible for avirulent pathogens-triggered changes in microbiomes. For this, the authors should test whether flg22 and heat-killed *P. syringae* also trigger similar changes. In addition, the authors should directly add avirulent pathogens to soil whether the enrichment of beneficial bacteria and promotion of plant growth and disease resistance are plant-dependent (cry-for-help). Related to this, did the authors find increased abundance of OTUs corresponding to DC3000 specifically in treated soil?
3. Root exudate data are not well connected with other observations. For instance, does adding myristic acid to soil cancel the avirulent strains-conditioned effects? This should happen if the hypothesis is true.
4. Disease resistance assays are not reliable or quantitative. The authors should measure CFU of DC3000. This is the standard method to measure bacterial disease resistance.

(Other comments)

1. In line 120, the definition of "nonhost pathogen" is not right.
2. The germination ratio in Fig 3B is not consistent with Fig 3D. In addition, the authors should

describe how many seeds they used and how they calculated.

3. In Fig 1A&C, the authors used 4-week-old *Arabidopsis* while in Fig 1B they measured the relative expression of PR1 in 15-day-old *Arabidopsis*. Why did they use different conditions?

4. In supplementary methods, the authors did not describe how and when they measured plant shoot fresh weight.

5. In supplementary methods, the authors should describe the mixture ratio of *Devosia* sp. CJK-A8-3 and *Devosia* sp. XL339.

6. Please provide the raw data of functional prediction by picrust2.

7. In the authors' previous paper (reference 25), they discussed JA and amino acids. They should connect findings in the previous and this study.

(Minor comments)

1. In line 102. *flhC* does not encode *flg22*.

2. In line 131. There is no total peak area in Fig 2C&D. The authors need to provide supporting data or change the statement.

3. It is better if the authors can show dots in figures like Fig 3C. And boxplots are more informative for readers.

4. In line 213, Fig 5F should be Fig 5HIJKL.

5. In line 293~240, two "however" sentences are stating the same thing and thus can be connected with "and".

6. Please add the legends about the values in Fig 2C&D and Fig 4C.

7. In Fig 3, the legend of C and D should be flipped.

Reviewer #2:

Remarks to the Author:

In this manuscript Liu and colleagues have performed a very interesting approach to show that avirulent DC3000 strains can protect host plants from future infections via modification of the rhizosphere microbiota. First, they test different bacterial mutants that can trigger different immune pathways and define a biotic stress. Then, they show that each treated plant has a different exudation profile, and, by performing conditioned-soils experiments, they show that the microbial communities are different depending on the strains with which the host was treated. Also, that those treated with avirulent strains protect the host from DC3000 infection. Isolation and validation of enriched bacterial strains shows that *Devosia* sp. could be one of the major targets in the host soil.

Overall, this study is very interesting and could have a strong impact in how to deal to with crop diseases in the future. Although it still remains to see whether this "vaccine" could work with other lab strains with also economical importance (for example, *Xcc*), this is a successful proof of concept. My major concern in this study comes with the omics data (metabolome and 16S), where only 3 replicates per condition were used. Although it seems enough for the authors to find interesting correlations and candidate bacterial OTUs, it is not sufficient for powerful statistical analysis and it is not the standard in the field. In addition, all of those conclusions are based in a single experiment, so I would strongly suggest to repeat these experiments to increase the sample size and re-do the statistical analysis. Thus, probably more bacterial strains will be enriched in the treatments of interest, which are possibly part of the strains you already isolated while trying to isolate *Devosia*.

P.S: the figure legends are of very bad quality but I guess it's because of some copy-pasting problem.

Some more specific comments:

- Introduction: changes in present and past tense are a bit confusing for the reader.

- Line 92-93: this sentence is unclear.

- Line 98-103: revise wording, here it seems D36E mutated itself. Could you maybe put in simple words what each line was meant to show (PTI vs ETI and vs no trigger at all)? You could refer to Figure 1E here.

- Figure 1B: there are no asterisks here, but letters. Please, modify the caption. Why is the untreated plant control not included here?

- Figure 1C: quantification of this phenotype?
- Line 114-117: I wouldn't say that you designed based on the experiment, because actually your selection is exactly the same as the initial experiment. I would probably start with the design of Figure 1E and then show your results, which justify that you use them for the rest of the experiments
- Figure 2: faulty labels in most of the panels,
- Figure 2B: DC3000 derivatives and wild type as well, no?
- How does the enrichment in Figure 4 differ to that of Figure 5?
- Because the big dispersion of samples in the PCoA of Figure 4B, I wanted to find the sequencing depth. But since I couldn't find the information, I summed the counts per sample in the OTU table and I was surprised to see that each sample has exactly the same number of reads (23849), which is a good depth, but very strange that all samples have the same. Did you do some filtering here?
- Line 227: p adjustment?

Methods:

- Col-0 seeds: stratified not vernalized
- Please, describe the scored hypersensitive response
- For root exudates collection: was the plant removed before? Was the agar attached to the roots included? All the devices for these measurements are in house?
- Enrichment tests: distribution of OTUs is usually not normal, and therefore a non-parametric test should be used instead. Also, check the last sentence of the paragraph for rhizosphere microbiome analysis, it says "genes whose abundance...". I guess it was meant to say OTUs. Please, provide more information about the Illumina Analysis Pipeline (what's the reference? It seems it processes samples even after sequencing?).
- What is MSgg medium?

Reviewer #3:

Remarks to the Author:

In recent years, it has been demonstrated that pathogen infection of leaves can lead to the changes in the composition of root microbiota that in turn provide protection against pathogen infection to the plant. In 2018, the authors demonstrated this "cry for help" response in *Arabidopsis* with the virulent bacterial pathogen *Pseudomonas syringae* pv. tomato DC3000. In the present study the authors moved on to investigate the effect of different PstDC3000 variants with different immune-eliciting capacities on this "cry for help" response. To this end they used PstDC3000 variants lacking ETI-eliciting effectors, lacking the PTI elicitor flagellin, and a variant ectopically expressing an effector (HopM1). Subsequently the authors investigated the effect of leaf infection by these PstDC3000 variants on the root microbiota, root exudate patterns, and effects of conditioned soils on next generation plant growth and defense. They identified a *Devosia* spp. that becomes enriched and when applied to the plants promotes growth and immunity.

Although I like the topic, research question, and the in-depth experimentation, I found this manuscript very difficult to read. Especially in the beginning where all kinds of well-established concepts in plant immunity research have been introduced and used in very confusing ways, just with the goal to demonstrate that the PstDC3000 variants used have different effects on activating the plant immune system. I had to read it several times before I could make sense of it. Very often the English grammar is not correct and strange words are used. This already starts with the title which is not correct English and is full of confusing wording. When looking through this, the bottom line of the work is that the authors demonstrate that different immune-eliciting variants of PstDC3000 affect root exudation and recruitment of microbiota with positive effects on plant growth and immunity. This was already demonstrated with the wild-type strain in 2018. The authors give the present work a different flavor by stressing that you don't need a virulent strain of a pathogen to induce the "cry for help" response, but that "avirulent" variants of a pathogen can do it as well. In this case the "avirulent" pathogens likely elicited to a large extent similar immune responses in the plant as did the wild-type strain (when looking at transcriptional profiles of *Arabidopsis* exposed to these different variants, one will see that the immune-related sectors of the transcriptome will largely overlap, but may vary in strength). Hence, in my view the authors only varied the strength of the elicited immune response in *Arabidopsis* with the different

PstDC3000 variants. It is thus not so surprising that many of the results are similar to those observed with the wildtype PstDC3000 strain, which although virulent also triggers immune responses in the plant. In sum, with the very difficult to follow first part of the work and the conceptually not so novel follow up part of the work, I feel that this manuscript is in its present form not suitable for a journal such as Nature Communications. Below my more detailed comments that are hopefully useful for the authors to improve their work for future submission.

1. The title is not correct English. Also the phrase "avirulent phytopathogen vaccine" is not correct and can even be misleading.
2. L.33: Use of the word "vaccine": The word "vaccine" is misplaced in the context of this study. I understand that the authors try to make a parallel with the human system, but it is far off. The "avirulent" pathogen variants used do not boost the immune system of the plant in the sense that we know it. And in the context of this study it recruits microbiota to the roots, a phenomenon that is not associated with vaccines. Hence, I would certainly not use this term at all. Moreover, we don't have to humanize the plant world in order to make it more appealing for the readers. It is of high importance already by itself.
3. L.39: The word "legatary soil" is very uncommon and as far as I know never used elsewhere. Many readers will not know what is meant by it. I suggest to rephrase it.
4. L.39: "Conclusively" is not the correct word here. I assume the authors mean "In conclusion". There are many places with English words that are not correctly used. I strongly advise the authors to have their manuscript checked by a native English speaker who also is able to understand the biology and biological terms used in the study.
5. L.41: The word "antibodies" is totally misplaced. I understand that the authors want to make a parallel with the human system, but plants don't have antibodies so this is misleading.
6. L.57: "Beans" should be "bean"
7. L.62: The authors suggest that using avirulent strains of a pathogen hold promise for sustainable agricultural development as they would elicit the cry for help response without causing disease. However, avirulent pathogens strongly elicit the immune system of the plant, so this will likely cause growth-defense trade offs and thus a reduction of yield. In my view this outlook is not very well thought through.
8. L.69: "MAMP" should be "MAMPs".
9. L.70: At many places the authors do correctly present the different elements of the plant immune system (PTI, ETI etc.). PTI does not prevent further infection, it only inhibits pathogen infection to a certain extent, this in contrast to ETI which leads to full resistance. It almost never leads to full resistance or full inhibition of pathogen growth. In the present study the authors don't see symptoms, but that doesn't mean that the pathogen is fully stopped. Be more correct in the description and align with consensus descriptions of the plant immune system.
10. L.78-79: Very unclear what is meant here. The wild-type virulent PstDC3000 doesn't suppress ETI, as in the Arabidopsis-Col-0 interaction studied here, Col-0 doesn't have R-genes that recognize one of the effectors of Pst DC3000 (hence no ETI elicited, so it can also not be suppressed).
11. L.80: "plant immunity" should be "plant immune system".
12. L.80: should be "elicitors for the "cry for help" response".
13. L.85: "trigger" should be "triggers". Check English throughout.
14. L.86: I also have difficulties with the word "avirulent". Classically, the word "avirulent" is used for pathogens that possess effectors (avirulence proteins) that are recognized by R-gene receptors, resulting in the activation of a hypersensitive response, an immune response we call ETI. Here the authors also use the word "avirulent" for Pst DC3000 mutant D36E, which doesn't have any effectors at all. Because of this Col-0 will have a stronger PTI response (as it is not suppressed by the effectors) resulting in enhanced resistance. It is confusing to call this mutant also "avirulent", as for readers with the classic terminology in mind this may not be able to fully follow it.
15. L.86-88: I think the comparison between "virulent" and "avirulent" variants of the pathogen are not so much different. The virulent pathogen induces a relatively mild immune response (sufficient to activate the "cry for help" response). The "avirulent" variants will stimulate a stronger immune response that is qualitatively largely similar. This explains why the virulent wildtype Pst DC3000 performs not so much different as the "avirulent" variants in terms of elicitation of changes in the root exudates and changes in the root microbiota.
16. L.91: "effectors" instead of "effector".

17. L.93: It would be more logically phrased if the authors would reason along the line of a gradient in the activation of plant immune responses, rather than stressing the presence or absence of effectors.
18. L.94: The observation that bacterial function is enriched for "secondary metabolism" is very vague and doesn't add much to our understanding of the phenomenon under study here.
19. L.97: A much better description of the system is needed. PstDC3000 is used on Arabidopsis accession Col-0 on which it is virulent. Hence, Col-0 does not have R genes that recognize the 36 effectors of the PstDC3000 strain used, so no ETI. ETI may only be triggered when effector HopM1 is introduced in the PstDC3000 strain. The authors mention that HopM1 leads to ETI in *Nicotiana benthamiana*, but the question is: does it trigger ETI in Arabidopsis accession Col-0? I'm sure there are reports about this. This should all be better explained, because as it now stands it is very unclear and inconsistent.
20. L.98: "Avirulent" is often used in the context of a variant that produces effectors that trigger ETI. In this case the D36E also seems to be avirulent but because of a different reason: no suppression of PTI due to the lack of effectors, so a stronger PTI response is triggered in these plants. As at the whole genome transcriptome level the PTI and ETI response largely overlap, both ways of inducing the immune system may lead to similar outcomes with respect to root exudate changes and recruitment of microbiota.
21. L.101: The wild-type Pst DC3000 is a virulent strain. It also triggers PTI, but to a lesser extent than D36E, because the latter doesn't have the PTI-suppressing effectors that Pst DC3000 has. So it is more a matter of magnitude. Stating that D36E was used as a PTI eliciting strain is thus not correct. Pst DC3000 itself is also a PTI eliciting strain.
22. L.102: very unclearly phrased sentence.
23. L.105: HopM1 was shown to trigger ETI in *Nicotiana benthamiana*. But here it is used on Arabidopsis Col-0. What is known about its ETI-eliciting capacity on Col-0? A picture with a bit more ROS production is not enough evidence I would say.
24. Figure 1A: strange why D36EFLC didn't cause disease. There is no Flg22-mediated PTI triggered so one would expect it to be virulent (weaker activation of PTI). Because of other MAMPs triggered PTI anyway?
25. Figure 1A: unclear picture. It seems that D36EHPM also shows symptoms like the wild type.
26. L.109: PR1 is not a resistance marker. It is a marker for the activation of a defence response. This does not necessarily lead to resistance.
27. L.115: It is not clear whether or not D36EHPM triggered a ETI response in Col-0. A little increase in ROS is not evidence.
28. L.117: The word "defeated" seems to be wrongly chosen here.
29. L.118: I would avoid using the word "avirulent" for D36E variants. Only D36EHPM may be avirulent in the classical sense, but then one needs to be sure that HopM1 elicits an HR in Col-0 (which is also not clearly shown).
30. L.120: "while ETI is a response to the nonhost pathogen". This is also very wrong. An avirulent variant of a pathogen is NOT a nonhost pathogen. The authors really have to more carefully use their wording and align with consensus terminology in the field.
31. Figure 1D: several observations do not fit with what one would expect. For instance: WT does trigger PTI, but likely weaker than D36E; D36EHPM should also show PTI, but that is possibly dampened by HopM1. In 1E it says that D36EFLC doesn't trigger PTI, but in 1D it is stated that it induces a weak one (as one would expect). So 1D and 1E do not completely align, which confuses the reader.
32. L.160: One would expect that D36EHPM would induce a soil-borne legacy that affects immunity, but it doesn't seem so in Figure 3E. Explanation?
33. L.171: It seems that wildtype Pst DC3000 doesn't have an effect, but this is different than in the 2018 publication. Explanation?
34. L.188: The observation that microbial functions relate to secondary metabolism is a very vague and general statement. It doesn't provide new insights into the phenomenon.
35. L.256: The use of the term "antibody" is wrong and misleading. I would avoid humanization of plant processes that by itself are very important.
36. L.270: Same for "vaccine" etc.

Remarks to the Author:

Comments

Plants are frequently attacked by diverse pathogens/pests, and many of them had been reported to trigger a plant mediated "cry for help" phenomenon which enriches disease suppression and growth promotion microbiota in the rhizosphere. Understanding how do plants cry for help to the microbiota is of giant application potential and increasing general interest. The manuscript by Liu et al., elegantly utilized a batch of pathogen mutants, which was believed to trigger different levels of biotic stresses to the host plant. The most important finding here is that infection by avirulent mutant of pathogen could still trigger the occurrence of "disease suppressive soil" without serious disease symptoms. This finding significantly furthered our understanding about the "cry for help" phenomenon, and has broad application potential in agricultural. The authors also characterized the key microbes recruited by upground pathogen infection, and further identified the key metabolic changes in the root exudate which might be related to the recruitment of disease suppressive microbes. Overall, the manuscript is carefully designed and well writtern, and would be of great interest to the broad readers in plant-microbe interaction field, and fit NC journal.

I still have several comments/concerns related to this work:

- 1) The term "vaccine" usually refers to dead bacterial or virus particles. Can the authors update the term to "attenuated live vaccine" or something like that?
- 2)the authors infiltrated "leaves of 15-day-old gnotobiotic Arabidopsis grown in MS medium with Pst DC3000 and the derived mutants". The leaves of 15-day-old gnotobiotic Arabidopsis should be very tiny and how can this be "infiltrated"? can we get a bit more description in the method. Also, for this system, is there a possibility that pathogen contamination in the rhizosphere, and they would utilize root exudate for proliferation. That will indirectly affects the composition of root exudates?
- 3)the authors systematically studied the changes in the root exudates upon infection, which is a critical part of the manuscript. However, it is not well concluded in the abstract part.
- 4)The authors did network analysis for the rhizosphere microbial community changes upon pathogen (mutants) infection (Fig. S6). However, more network related parameters (connectance, module numbers, degree centralization.....) could be listed to show the overall features of networks.
- 5)In the introduction part (line 65), a lot recent exciting breakthrough which supports that root exudate could shape rhizosphere microbiota should be briefly introduced here (following the root exudate part). Like : Yu, Peng, et al. "Plant flavones enrich rhizosphere Oxalobacteraceae to improve maize performance under nitrogen deprivation." *Nature plants* 7.4 (2021): 481-499. ; Huang A C, Jiang T, Liu Y X, et al. A specialized metabolic network selectively modulates Arabidopsis root microbiota[J]. *Science*, 2019, 364(6440): eaau6389.

Response to reviewers' comments

Reviewer #1 (Remarks to the Author):

This manuscript reports that infection with avirulent strains of the bacterial pathogen *Pseudomonas syringae* DC3000 leads to the enrichment of beneficial soil microbes that promote growth and disease resistance of *Arabidopsis thaliana*. The authors identified OTUs belonging to the genus *Devosia* that can promote these plant traits. Because of this, they propose that avirulent pathogens can use as plant vaccines.

The authors first characterized *Arabidopsis* immune responses to different strains of *P. syringae* DC3000 including wild-type, type III effector-deficient (D36E), Flc-deficient in addition to D36E (D36EFLC), and D36E carrying an effector HopM1 (D36EHPM). Then, they performed three rounds of leaf treatment with these strains, followed by measurement of root exudates, shoot fresh weight, and resistance against wild-type *P. syringae* DC3000. They observed that in particular, D36E and D36EFLC conditioned soil microbes promoted plant growth and resistance. They identified differentially abundant OTUs and isolated bacterial strains corresponding to these OTUs. Strains belonging to *Devosia* OTUs could recapitulate effects of D36E and D36EFLC-conditioned soil on plant growth and disease resistance promotion.

Pathogen-infection-triggered enrichment of beneficial bacteria that promote disease resistance has been well documented including *P. syringae* (Berg and Koskella *Curr Biol* 2018; Yuan et al *Microbiome* 2018). Thus, the novelty in this manuscript is that this can also happen by avirulent pathogens. Identification of OTUs belonging to *Devosia* for promoting growth and resistance is also a novel finding of this paper.

While this study adds to our understanding of “cry-for-help”, it remains unclear how avirulent pathogens do this. To this end, it would be highly informative if the authors perform a series of experiments using immune elicitors such as flg22. This answers questions about whether plant immune activation by avirulent (or virulent) pathogens is responsible for cry-for-help or not. Related to this, it remains unclear whether avirulent strains promote the enrichment of beneficial bacteria directly via bacteria-bacteria interactions or through plant immune activation or both.

In summary, their findings contribute to the research community, but the mechanistic depth should be improved. In addition, there are points that the authors should consider to improve this study.

Response: Thank you for your overall comments. Your question of how avirulent pathogen trigger plant “cry for help” response is very insightful. The results of the original manuscript suggest flg22 should not play an essential role in triggering “cry for help” response (comparing D36E and D36EFLC), so we think only testing flg22 in those experiment cannot fully answer the question. Since D36EFLC, causing very weak PTI, triggered plant “cry for help”, we hypothesize that immune response may

not essential and some other metabolite produced by this strain may be responsible for the activation. We tested the metabolites of D36E and D36EFLC in absence of alive bacterial cell and demonstrated they are sufficient to trigger plant “cry for help” response. Interestingly, we also found D36EHPM could quench this effect, suggesting that ETI may suppress the “cry for help” response at a higher priority level. We believe these additional results provide deep insights about the mechanisms of “cry for help”. For the question of “whether avirulent strains promote the enrichment of beneficial bacteria directly via bacteria-bacteria interactions or through plant immune activation or both”, DC3000 was inoculated to the leave without contact with the rhizosphere microbiome, so the promotion of the enrichment of beneficial bacteria in rhizosphere via the bacteria-bacteria interaction can be excluded, to confirm this, we searched the microbiome data for 16S rDNA sequence belong to DC3000 and found no hit. All other comments have been addressed, please find the detailed responses below. We hope you will be satisfied with the revised manuscript.

(Major comments)

1. DC3000, which contains HopM1, does not trigger ETI in Arabidopsis. Why does D36EHPM trigger ETI? There is no evidence that HopM1 triggers ETI in Arabidopsis (the authors need to find the receptor for HopM1 in Arabidopsis). Furthermore, D36EFLC and DC3000 also trigger PTI. Thus, it is strange to categorize these strains as in Fig1. Defining threat level by measuring plant immune responses does not make sense. If the authors really want to define it, they can measure fitness affected by infection with different strains.

Response: Thank you for your comment. First, D36EHPM triggers ETI in Arabidopsis while wild type DC3000 does not, this is possible when the effector in wild-type DC3000 that responsible for suppressing the HopM1 recognition by Arabidopsis receptor was absent in D36E background, which is very common in interaction between DC3000 and plant [1]. Although the receptor for HopM1 in Arabidopsis is still to be identified, the effectors that suppress the perception of HopM1 by *N. benthamiana* receptor has been identified (AvrPtoB_{M3} and HopI1 [1]), it suggests that suppressing ETI by another effector is a very common strategy for a successful pathogen. In the revised manuscript, we provided the images shown the hypersensitive response elicited by D36EHPM, which indicate D36EHPM could trigger ETI in Arabidopsis. Second, we are sorry for the incorrect description of these strains. D36EFLC and DC3000 also trigger weaker PTI. We have rewritten the section without defining the threat levels.

[1] Wei, H.L. *et al.* (2018) Modular Study of the Type III Effector Repertoire in *Pseudomonas syringae* pv. tomato DC3000 Reveals a Matrix of Effector Interplay in Pathogenesis. *Cell Rep.* 23, 1630–1638

2. As mentioned above, it remains unclear what is responsible for avirulent pathogens-triggered changes in microbiomes. For this, the authors should test whether flg22 and heat-killed *P. syringae* also trigger similar changes. In addition, the authors should directly add avirulent pathogens to soil whether the enrichment of

beneficial bacteria and promotion of plant growth and disease resistance are plant-dependent (cry-for-help). Related to this, did the authors find increased abundance of OTUs corresponding to DC3000 specifically in treated soil?

Response: Thank you for your comment. First, in the original manuscript, we have included the mutant D36EFLC in this study, which is deficient in flg22 synthesis, this mutant still induced the beneficial effect. This result can exclude the essential function of flg22 to induce plant "cry for help" response. So, we did not test flg22 for trigger the "cry for help". Following your idea that heat-killed *P. syringae* may trigger plant "cry for help" response, we further expanded your hypothesis that some other secreted signal molecules of D36E and D36EFLC, rather than the alive bacterial cells themselves, may be responsible for activating "cry for help", we also suppose that D36EHPM triggered ETI would block the effect because our data in the original manuscript showed that D36EHPM did not trigger plant "cry for help" response. In the revised manuscript, we treated *Arabidopsis* leaves with culture supernatant of D36E or D36EFLC, and inoculated leaves with or without D36EHPM in the meantime to influence the soil microbiome. These treatments were applied for three generations for enrichment of soil-borne legacy. As expected, the culture supernatant induced a soil-borne legacy that promote plant growth and suppress disease, but this induction was blocked when alive D36EHPM strain was co-inoculated. These additional results have been added to the revised manuscript. We believe these results would provide new insights for understanding the molecule(s) that responsible for pathogen triggered changes in microbiomes.

Second, because the nonpathogenic derivatives of DC3000 were inoculated on leaves without direct no contact of the soil microbiome, so the enrichment of beneficial bacterial and promotion of plant growth and disease resistance are plant dependent. Additionally, we have searched the raw data of rhizosphere microbiome for the 16S rDNA sequence belong to DC3000 or the derivatives and found no increased abundance of OTUs corresponding to DC3000 (no hits for all samples), indicating the inoculation has not contaminated the soil, this finding support that the effect of leaf-inoculation on the enriched legacy is plant dependent. We have included this information in the revised manuscript.

3. Root exudate data are not well connected with other observations. For instance, does adding myristic acid to soil cancel the avirulent strains-conditioned effects? This should happen if the hypothesis is true.

Response: Thank you for your valuable suggestion. We performed an experiment adding myristic acid to the soil with the disease-suppressive and growth promotive legacy. Using SL-CK, SL-D36E and SL-D36EFLC ("SL" refer to the soil contains the legacy) as control, we found adding myristic acid to SL-D36E and SL-D36EFLC did **significantly reduced** the conditioned effects to promote plant growth and suppress disease, suggesting that reduced root exudation of myristic acid contribute to the enrichment of beneficial soil-borne legacy. This result has been added to the revised manuscript.

4. Disease resistance assays are not reliable or quantitative. The authors should measure CFU of DC3000. This is the standard method to measure bacterial disease resistance.

Response: Thank you for your suggestion. We reformed this experiment with the preserved conditioned soil in previous experiment and measured the CFU of DC3000 by plate counting. The results are consistent with previous conclusion. This result has been supplied and the manuscript was revised accordingly. This method was also used for evaluation of the disease resistance in the additional experiments.

(Other comments)

1. In line 120, the definition of “nonhost pathogen” is not right.

Response: Thank you for your comment. We have rewritten this section to avoid using “nonhost pathogen”.

2. The germination ratio in Fig 3B is not consistent with Fig 3D. In addition, the authors should describe how many seeds they used and how they calculated.

Response: We are sorry for our mistakes that misleading you. In Fig. 3B, five seeds were sowed to each pot and the germination ratio was recorded (20 pots) for each pot for further analyzing the average germination ratio in each pot. So, the results shown in Fig. 3B is consistent with the image shown in Fig. 3D. We have carefully revised the description in methods, results and legends for clear.

3. In Fig 1A&C, the authors used 4-week-old Arabidopsis while in Fig 1B they measured the relative expression of PR1 in 15-day-old Arabidopsis. Why did they use different conditions?

Response: Both tests were performed with 4-week-old Arabidopsis seedlings. It has been corrected. We are sorry for this mistake and have thoroughly checked and corrected the methods.

4. In supplementary methods, the authors did not describe how and when they measured plant shoot fresh weight.

Response: Thank you for your comment. The plant shoot fresh weight were measured at 5-week post sowing by weighing the aboveground tissue directly. We have added this information in the revised manuscript.

5. In supplementary methods, the authors should describe the mixture ratio of *Devosia* sp. CJK-A8-3 and *Devosia* sp. XL339.

Response: Thank you for your suggestion. *Devosia* sp. CJK-A8-3 and *Devosia* sp. XL339 are equally mixed. This information has been added in methods.

6. Please provide the raw data of functional prediction by picrust2.

Response: Thank you for your comment. Following reviewers #3' comment 18 and comment 34, we believe the functional prediction of the microbiome seem

insignificant to the objective of this study. So, we removed this result in the revised manuscript.

7. In the authors' previous paper (reference 25), they discussed JA and amino acids. They should connect findings in the previous and this study.

Response: Thank you for your comment. Following your suggestion, we have compared the relative content of amino acids in Fig. 2, it showed that amino acids were increased in root exudates of wild-type DC3000 infected Arabidopsis, which is consistent with previous paper [2]. This result has been added into the root exudates analysis section (Fig. 2). We also discussed JA to connect with the previous findings.

[2] Yuan, J. *et al.* (2018) Root exudates drive the soil-borne legacy of aboveground pathogen infection. *Microbiome* 6, 156

(Minor comments)

1. In line 102. fliC does not encode flg22.

Response: Thank you for your comment. The sentence has been corrected.

2. In line 131. There is no total peak area in Fig 2C&D. The authors need to provide supporting data or change the statement.

Response: Thank you for your comment. It has been revised to "relative content".

3. It is better if the authors can show dots in figures like Fig 3C. And boxplots are more informative for readers.

Response: Thank you for your comment. We have changed the figure to boxplots and have shown dots for all the tested data for plant assay.

4. In line 213, Fig 5F should be Fig 5HIJKL.

Response: Sorry for the mistake. The label has been revised.

5. In line 293~240, two "however" sentences are stating the same thing and thus can be connected with "and".

Response: Thank you for your comment. These sentences have been revised following your suggestion.

6. Please add the legends about the values in Fig 2C&D and Fig 4C.

Response: Thank you for your comment. The legend about the values has been added.

7. In Fig 3, the legend of C and D should be flipped.

Response: Sorry for the mistake. This legend has been revised.

Reviewer #2 (Remarks to the Author):

In this manuscript Liu and colleagues have performed a very interesting approach to show that avirulent DC3000 strains can protect host plants from future infections via modification of the rhizosphere microbiota. First, they test different bacterial mutants that can trigger different immune pathways and define a biotic stress. Then, they show that each treated plant has a different exudation profile, and, by performing conditioned-soils experiments, they show that the microbial communities are different depending on the strains with which the host was treated. Also, that those treated with avirulent strains protect the host from DC3000 infection. Isolation and validation of enriched bacterial strains shows that *Devosia* sp. could be one of the major targets in the host soil.

Overall, this study is very interesting and could have a strong impact in how to deal with crop diseases in the future. Although it still remains to see whether this “vaccine” could work with other lab strains with also economical importance (for example, Xcc), this is a successful proof of concept. My major concern in this study comes with the omics data (metabolome and 16S), where only 3 replicates per condition were used. Although it seems enough for the authors to find interesting correlations and candidate bacterial OTUs, it is not sufficient for powerful statistical analysis and it is not the standard in the field. In addition, all of those conclusions are based in a single experiment, so I would strongly suggest to repeat these experiments to increase the sample size and re-do the statistical analysis. Thus, probably more bacterial strains will be enriched in the treatments of interest, which are possibly part of the strains you already isolated while trying to isolate *Devosia*.

P.S: the figure legends are of very bad quality but I guess it's because of some copy-pasting problem.

Response: Thank you for your positive comment. In fact, in this study, we have repeated the experiment twice with similar results, and we have 6 replicates for the rhizosphere soil DNA for each treatment, considering that our experiment was conducted with pots in the well-controlled greenhouse, and all these replicates are well consistent. Previously, we only randomly selected three samples for sequencing and further analysis. In the revised manuscript, we sequenced the other 3 replicated samples and, included them in further statistical analysis. We also repeated the root exudates collection and analysis with 6 replicates for each treatment. Moreover, other additional experiments suggested by other reviewers were also performed with 6 replicates. We have redone the statistics with the new data, which showed clear trends in PCoA and consistently revealed the outstanding role of *Devosia* and myristic acid in the “cry for help” response. The manuscript was revised accordingly.

We are sorry for the bad figure legends; we have carefully revised the legend to be more informative. We believe the revised manuscript have been improved following your advice.

Some more specific comments:

- Introduction: changes in present and past tense are a bit confusing for the reader.

Response: We have had our manuscript edited by the professional language editing service of Springer Nature to polish the language.

- Line 92-93: this sentence is unclear.

Response: Thank you for your comment. The sentence has been revised to “We demonstrate that two nonpathogenic derivatives of Pst DC3000 can induce plants to establish the disease suppressive soil-borne legacy”.

- Line 98-103: revise wording, here it seems D36E mutated itself. Could you maybe put in simple words what each line was meant to show (PTI vs ETI and vs no trigger at all)? You could refer to Figure 1E here.

Response: We have rewritten this section for clarity. Please find the detailed description in the first part of the result.

- Figure 1B: there are no asterisks here, but letters. Please, modify the caption. Why is the untreated plant control not included here?

Response: Sorry for the mistake. The description of the statistics was revised. The untreated plant control was used for normalization (the value is zero if shown in this figure).

- Figure 1C: quantification of this phenotype?

Response: Thank you for your comment. We have quantitatively measured the ROS by using L-012 probe.

- Line 114-117: I wouldn't say that you designed based on the experiment, because actually your selection is exactly the same as the initial experiment. I would probably start with the design of Figure 1E and then show your results, which justify that you use them for the rest of the experiments

Response: Thank you for your comment. We have removed these confusing sentences.

- Figure 2: faulty labels in most of the panels,

Response: Thank you for your comment. Figure 2 has been replotted and revised to avoid faulty labels.

- Figure 2B: DC3000 derivatives and wild type as well, no?

Response: Sorry for the confusing descriptions. It was revised to “wild type DC3000 treated- and DC3000 derivative-treated plants”.

- How does the enrichment in Figure 4 differ to that of Figure 5?

Response: Sorry for the confusing figures. Figure 4a can only show the enriched bacterial group for one treatment. In Figure 5, we identified the significantly enriched bacterial groups in both D36E and D36EFLC treatments. However, considering that the original figure 4a was not necessary for the main goal and the large sets of results

added in the revised manuscript, we removed this panel and combined the original Figure 4 with the original Figure 5 to generate the current Figure 4 in the revised manuscript.

- Because the big dispersion of samples in the PCoA of Figure4B, I wanted to find the sequencing depth. But since I couldn't find the information, I summed the counts per sample in the OTU table and I was surprised to see that each sample has exactly the same number of reads (23849), which is a good depth, but very strange that all samples have the same. Did you do some filtering here?

Response: Thank you for your comments. First, we are sorry for missing the detailed description of the data filtering. In previous version, the OTU table we used for diversity and composition analysis was a rarefied OTU table. Constructing a rarefied OTU table, in which the reads of each sample are rarefied to the same abundance (usually based on the sample with the fewest reads), is a standard protocol on QIIME. Besides, it has been proved that rarefying itself does not increase the false discovery rate if the average library size for each group is approximately equal, and rarefying may help decrease the false discovery rate when large (~10x) differences occur in the average library size between groups [3]. In the revised manuscript, we rarefied all samples to have the same read counts of 15000. We have clearly described the analysis procedure in the revised manuscript.

Second, we have extended the replicates to 6 and regenerated the PCoA, in which the samples are not that dispersed.

[3] Weiss, S. *et al.* (2017) Normalization and microbial differential abundance strategies depend upon data characteristics. *Microbiome* 5, 1–18

- Line 227: p adjustment?

Response: Sorry for the confusing words. We meant q value here. It has been revised.

Methods:

- Col-0 seeds: stratified not vernalized

Response: Thank you for your comment. The word has been corrected.

- Please, describe the scored hypersensitive response

Response: Thank you for your comment. We have provided clear image for the observed hypersensitive response caused by D36EHPM and included the scored hypersensitive response above the images.

- For root exudates collection: was the plant removed before? Was the agar attached to the roots included? All the devices for these measurements are in house?

Response: Thank you for your comment. The plant was removed before collecting the agar containing the root exudates, the agar attached to the roots were also included. All the collection processes were performed in a growth chamber. This information has been added in the revised method.

- Enrichment tests: distribution of OTUs is usually not normal, and therefore a non-parametric test should be used instead. Also, check the last sentence of the paragraph for rhizosphere microbiome analysis, it says “genes whose abundance...”. I guess it was meant to say OTUs. Please, provide more information about the Illumina Analysis Pipeline (what’s the reference? It seems it processes samples even after sequencing?).

Response: Thank you for your comments. First, following your suggestion, we changed the method of statistical analysis for distribution of OTUs to non-parametric Kruskal-Wallis test. Second, sorry for the confusing description, we meant OTU here. It has been revised for clear. Third, following your suggestion, we provided the detailed information about the analysis pipeline and the references. Please find the description in the method.

- What is MSgg medium?

Response: Sorry for missing the description of the medium. It is a medium used for biofilm formation test. We have added detailed description for this medium in both results and method parts.

Reviewer #3 (Remarks to the Author):

In recent years, it has been demonstrated that pathogen infection of leaves can lead to the changes in the composition of root microbiota that in turn provide protection against pathogen infection to the plant. In 2018, the authors demonstrated this “cry for help” response in Arabidopsis with the virulent bacterial pathogen *Pseudomonas syringae* pv. tomato DC3000. In the present study the authors moved on to investigate the effect of different PstDC3000 variants with different immune-eliciting capacities on this “cry for help” response. To this end they used PstDC3000 variants lacking ETI-eliciting effectors, lacking the PTI elicitor flagellin, and a variant ectopically expressing an effector (HopM1). Subsequently the authors investigated the effect of leaf infection by these PstDC3000 variants on the root microbiota, root exudate patterns, and effects of conditioned soils on next generation plant growth and defense. They identified a *Devosia* spp. that becomes enriched and when applied to the plants promotes growth and immunity.

Although I like the topic, research question, and the in-depth experimentation, I found this manuscript very difficult to read. Especially in the beginning where all kinds of well-established concepts in plant immunity research have been introduced and used in very confusing ways, just with the goal to demonstrate that the PstDC3000 variants used have different effects on activating the plant immune system. I had to read it several times before I could make sense of it. Very often the English grammar is not correct and strange words are used. This already starts with the title which is not correct English and is full of confusing wording. When looking through this, the bottom line of the work is that the authors demonstrate that different immune-eliciting variants of PstDC3000 affect root exudation and recruitment of microbiota with positive effects on plant growth and immunity. This was already demonstrated with the wild-type

strain in 2018. The authors give the present work a different flavor by stressing that you don't need a virulent strain of a pathogen to induce the "cry for help" response, but that "avirulent" variants of a pathogen can do it as well. In this case the "avirulent" pathogens likely elicited to a large extent similar immune responses in the plant as did the wild-type strain (when looking at transcriptional profiles of Arabidopsis exposed to these different variants, one will see that the immune-related sectors of the transcriptome will largely overlap, but may vary in strength). Hence, in my view the authors only varied the strength of the elicited immune response in Arabidopsis with the different PstDC3000 variants. It is thus not so surprising that many of the results are similar to those observed with the wildtype PstDC3000 strain, which although virulent also triggers immune responses in the plant. In sum, with the very difficult to follow first part of the work and the conceptually not so novel follow up part of the work, I feel that this manuscript is in its present form not suitable for a journal such as Nature Communications. Below my more detailed comments that are hopefully useful for the authors to improve their work for future submission.

Response: Thank you for your overall comments and we are very sorry for the bad readability and the confusing descriptions that made it difficult to understand. First, we have carefully revised the manuscript according to your professional advices to make the manuscript clear, especially in the description of the first part of the results.

Second, we have dropped the idea to over define the strains and their elicited responses. As you suggested, we only used the strains to elicit different levels of plant immune response.

Finally, we appreciated your detailed advice to the writing and descriptions, which did help a lot to improve the quality of the manuscript. We have revised the manuscript according to your comments and hope you will be satisfied with the current version.

1. The title is not correct English. Also the phrase "avirulent phytopathogen vaccine" is not correct and can even be misleading.

Response: The title has been revised to "Nonpathogenic mutants of phytopathogen trigger the plant "cry for help" response to assembly the beneficial rhizomicrobiome".

2. L.33: Use of the word "vaccine": The word "vaccine" is misplaced in the context of this study. I understand that the authors try to make a parallel with the human system, but it is far off. The "avirulent" pathogen variants used do not boost the immune system of the plant in the sense that we know it. And in the context of this study it recruits microbiota to the roots, a phenomenon that is not associated with vaccines. Hence, I would certainly not use this term at all. Moreover, we don't have to humanize the plant world in order to make it more appealing for the readers. It is of high importance already by itself.

Response: Thank you for your suggestion. We realize that making a parallel with the human system is not suitable here. We have revised the manuscript to avoid using the word "vaccine", "antibody" and humanizing the plant.

3. L.39: The word “legatary soil” is very uncommon and as far as I know never used elsewhere. Many readers will not know what is meant by it. I suggest to rephrase it.

Response: Thank you for your comment. The word “legatary soil” has been revised to “soil containing the soil-borne legacy” with abbreviation as “SL”.

4. L.39: “Conclusively” is not the correct word here. I assume the authors mean “In conclusion”. There are many places with English words that are not correctly used. I strongly advice the authors to have their manuscript checked by a native English speaker who also is able to understand the biology and biological terms used in the study.

Response: Sorry for the confusing wording. The “Conclusively” has been revised to “In conclusion”. We have had our manuscript edited by the professional language editing service of Springer Nature to polish the language.

5. L.41: The word “antibodies” is totally misplaced. I understand that the authors want to make a parallel with the human system, but plants don’t have antibodies so this is misleading.

Response: Thank you for your comment. The word “antibody” “antibodies” have been removed.

6. L.57: “Beans” should be “bean”

Response: Thank you for your comment. The word has been corrected.

7. L.62: The authors suggest that using avirulent strains of a pathogen hold promise for sustainable agricultural development as they would elicit the cry for help response without causing disease. However, avirulent pathogens strongly elicit the immune system of the plant, so this will likely cause growth-defense trade offs and thus a reduction of yield. In my view this outlook is not very well thought through.

Response: Thank you for your comment. First, we have revised the word “avirulent” to “nonpathogenic” following your suggestion. We demonstrate using the nonpathogenic derivatives D36E and D36EFLC can trigger plant “cry for help” response. Second, we agree with you that using a nonpathogenic derivative of DC3000 such as D36EFLC still elicit weak immune response, it should not be a final solution for eliciting the “cry for help” response due to the growth-defense trade-offs. But our finding demonstrates the possibility to find an exact plant-friendly molecule that trigger the “cry for help” response.

Moreover, in the revised manuscript, we found the supernatant of D36EFLC culture could elicit the “cry for help” responses. We believe these new results will contribute to the possible application. Even though, we understand the current results are still insufficient for direct field application, so we have revised in discussion to tune down the suggestions.

8. L.69: “MAMP” should be “MAMPs”.

Response: Thank you for your comment. The word has been corrected.

9. L.70: At many places the authors do correctly present the different elements of the plant immune system (PTI, ETI etc.). PTI does not prevent further infection, in only inhibits pathogen infection to a certain extent, this in contrast to ETI which leads to full resistance. It almost never leads to full resistance or full inhibition of pathogen growth. In the present study the authors don't see symptoms, but that doesn't mean that the pathogen is fully stopped. Be more correct in the description and align with consensus descriptions of the plant immune system.

Response: Thank you for your comment. The sentence has been revised to "Recognition of MAMPs elicits pattern-triggered immunity (PTI) that inhibits pathogen infection".

10. L.78-79: Very unclear what is meant here. The wild-type virulent PstDC3000 doesn't suppress ETI, as in the Arabidopsis-Col-0 interaction studied here, Col-0 doesn't have R-genes that recognize one of the effectors of Pst DC3000 (hence no ETI elicited, so it can also not be suppressed).

Response: Thank you for your comment. We have removed the confusing descriptions in this part.

11. L.80: "plant immunity" should be "plant immune system".

Response: Thank you for your comment. The "plant immunity" has been corrected to "plant immune system".

12. L.80: should be "elicitors for the "cry for help" response".

Response: Thank you for your comment. The word has been corrected.

13. L.85: "trigger" should be "triggers". Check English throughout.

Response: Thank you for your comment. The word has been corrected. The English has been checked throughout the manuscript.

14. L.86: I also have difficulties with the word "avirulent". Classically, the word "avirulent" is used for pathogens that possess effectors (avirulence proteins) that are recognized by R-gene receptors, resulting in the activation of a hypersensitive response, an immune response we call ETI. Here the authors also use the word "avirulent" for Pst DC3000 mutant D36E, which doesn't have any effectors at all. Because of this Col-0 will have a stronger PTI response (as it is not suppressed by the effectors) resulting in enhanced resistance. It is confusing to call this mutant also "avirulent", as for readers with the classic terminology in mind this may not be able to fully follow it.

Response: Thank you for your comment. We have revised the manuscript to avoid using the word "avirulent". We have uniformly rephrased the "avirulent" to "nonpathogenic"..

15. L.86-88: I think the comparison between “virulent” and “avirulent” variants of the pathogen are not so much different. The virulent pathogen induces a relatively mild immune response (sufficient to activate the “cry for help” response). The “avirulent” variants will stimulate a stronger immune response that is qualitatively largely similar. This explains why the virulent wildtype Pst DC3000 performs not so much different as the “avirulent” variants in terms of elicitation of changes in the root exudates and changes in the root microbiota.

Response: Thank you for your comment. We have revised the sentence in introduction to be focused on comparing the effect of different level of plant immune response, rather than highlighting the comparison between virulent and “avirulent” (nonpathogenic).

16. L.91: “effectors” instead of “effector”.

Response: Thank you for your comment. The word has been corrected.

17. L.93: It would be more logically phrased if the authors would reason along the line of a gradient in the activation of plant immune responses, rather than stressing the presence or absence of effectors.

Response: Thank you for your comment. We have revised the sentence to not stressing the effectors, and only focus on the difference in activating plant immune responses.

18. L.94: The observation that bacterial function is enriched for “secondary metabolism” is very vague and doesn’t add much to our understanding of the phenomenon under study here.

Response: Thank you for your comment. We have removed this statement in this section.

19. L.97: A much better description of the system is needed. PstDC3000 is used on Arabidopsis accession Col-0 on which it is virulent. Hence, Col-0 does not have R genes that recognize the 36 effectors of the PstDC3000 strain used, so no ETI. ETI may only be triggered when effector HopM1 is introduced in the PstDC3000 strain. The authors mention that HopM1 leads to ETI in *Nicotiana benthamiana*, but the question is: does it trigger ETI in Arabidopsis accession Col-0? I'm sure there are reports about this. This should all be better explained, because as it now stands it is very unclear and inconsistent.

Response: Thank you for your comment. HopM1 mutation in background of D36E elicits ETI in Arabidopsis and *Nicotiana benthamiana*. The Arabidopsis ETI eliciting capacity of D36E with HopM1 has not been reported but has been confirmed by this study with robust data. We provided clearer image to show that infiltration of D36EHPM led clear HR in Arabidopsis leaves. We have added this result, please find the detailed data in Figure 1.

20. L.98: "Avirulent" is often used in the context of a variant that produces effectors that trigger ETI. In this case the D36E also seems to be avirulent but because of a different reason: no suppression of PTI due to the lack of effectors, so a stronger PTI response is triggered in these plants. As at the whole genome transcriptome level the PTI and ETI response largely overlap, both ways of inducing the immune system may lead to similar outcomes with respect to root exudate changes and recruitment of microbiota.

Response: Thank you for your comment. The word "avirulent" has been revised to "nonpathogenic" when describing D36E and D36EFLC throughout the text.

21. L.101: The wild-type Pst DC3000 is a virulent strain. It also triggers PTI, but to a lesser extent than D36E, because the latter doesn't have the PTI-suppressing effectors that Pst DC3000 has. So it is more a matter of magnitude. Stating that D36E was used as a PTI eliciting strain is thus not correct. Pst DC3000 itself is also a PTI eliciting strain.

Response: Thank you for your comment. We have revised the sentence and describing D36E as a "strong PTI-eliciting strain".

22. L.102: very unclear phrased sentence.

Response: Thank you for your comment. We have revised the sentence to "The *Pst* DC3000-derived nonpathogenic mutant D36E, in which the T3SS remains intact, but all 36 type III effectors are deleted, was used as a strong PTI-eliciting strain."

23. L.105: HopM1 was shown to trigger ETI in *Nicotiana benthamiana*. But here it is used on *Arabidopsis Col-0*. What is known about its ETI-eliciting capacity on *Col-0*? A picture with a bit more ROS production is not enough evidence I would say.

Response: Thank you for your comment. We have supplied images showing HR in *Arabidopsis* leaves caused by D36EHPM.

24. Figure 1A: strange why D36EFLC didn't cause disease. There is no Flg22-mediated PTI triggered so one would expect it to be virulent (weaker activation of PTI). Because of other MAMPs triggered PTI anyway?

Response: Thank you for your comment. D36EFLC did not cause disease because all of the effectors that essential for establish disease is deleted, while some weak PTI could still be activated by other MAMPs such as pilus.

25. Figure 1A: unclear picture. It seems that D36EHPM also shows symptoms like the wild type.

Response: Thank you for your comment. We have shown clearer image to show the symptom in *Arabidopsis* leaves.

26. L.109: PR1 is not a resistance marker. It is a marker for the activation of a defence response. This does not necessarily lead to resistance.

Response: Thank you for your comment. We have revised the phrases describing PR1.

27. L.115: It is not clear whether or not D36EHPM triggered a ETI response in Col-0. A little increase in ROS is not evidence.

Response: Thank you for your comment. We have supplied a clear image show the HR caused by D36EHPM.

28. L.117: The word “defeated” seems to be wrongly chosen here.

Response: Thank you for your comment. This sentence has been removed.

29. L.118: I would avoid using the word “avirulent” for D36E variants. Only D36EHPM may be avirulent in the classical sense, but then one need to be sure that HopM1 elicits an HR in Col-0 (which is also not clearly shown).

Response: Thank you for your comment. We have revised the word “avirulent” to “nonpathogenic”.

30. L.120: “while ETI is a response to the nonhost pathogen”. This is also very wrong. An avirulent variant of a pathogen is NOT a nonhost pathogen. The authors really have to more carefully use their wording and align with consensus terminology in the field.

Response: Thank you for your comment. We have removed this wrong sentence.

31. Figure 1D: several observations do not fit with what one would expect. For instance: WT does trigger PTI, but likely weaker than D36E; D36EHPM should also show PTI, but that is possibly dampened by HopM1. In 1E it says that D36EFLC doesn't trigger PTI, but in 1D it is stated that is induces a weak one (as one would expect). So 1D and 1E do not completely align, which confuses the reader.

Response: Thank you for your comment. Sorry for bad conclusion shown in the figure, we have carefully revised the conclusion in Figure 1 to fit the results of symptom.

32. L.160: One would expect that D36EHPM would induce a soil-borne legacy that affects immunity, but is doesn't seem so in Figure 3E. Explanation?

Response: Thank you for your comment. It seems that ETI may block the DC3000 induced soil-borne legacy that affect immunity. We performed additional experiments to test whether D36EHPM inoculation can quench the effect of D36E or D36EFLC on soi-borne legacy. The result showed that 1) treating plant leaves with supernatant of D36E (D36E/S) and D36EFLC (D36EFLC/S) can elicit the soil-borne legacy with disease-suppressing property; 2) inoculation with D36EHPM has quenched the effect of D36EFLC/S and D36E/S on eliciting disease suppressing property of the soil-borne legacy. These results demonstrated that metabolite of DC3000 could elicit plant “cry for help” response, while D36EHPM could block this elicitation. As it is generally agreed that plant cry for help when it does need help, while ETI is a stronger immune

response that completely prevent further infection, we can expect that plant does not need “cry for help”.

33. L.171: It seems that wild type Pst DC3000 doesn't have an effect, but this is different than in the 2018 publication. Explanation?

Response: Thank you for your comment. The PCoA differences between treatments can be covered sometimes. 1) In the PCoA for microbiome of rhizosphere soil presented by Yuan et al. [2], “pathogen-conditioned” and “control” are very close (in both main figure and supplementary figure of that paper), although with significant difference. 2) We have redone the sequencing to extend the replicate number to 6, the re-analysis with 6 replicates showed that WT and CK are clearly separated.

[2] Yuan, J. *et al.* (2018) Root exudates drive the soil-borne legacy of aboveground pathogen infection. *Microbiome* 6, 156

34. L.188: The observation that microbial functions relate to secondary metabolism is a very vague and general statement. It doesn't provide new insights into the phenomenon.

Response: Thank you for your comment. In agree with you, we feel the functional prediction is too descriptive and contribute less to the main statement of the manuscript, so we removed this part in the revised manuscript.

35. L256: The use of the term “antibody” is wrong and misleading. I would avoid humanization of plant processes that by itself are very important.

Response: Thank you for your comment. The words and the sentence that humanize of plant processes have been revised.

36. L.270: Same for “vaccine” etc.

Response: Thank you for your comment. The word “vaccine” has been revised.

Reviewer #4 (Remarks to the Author):

Comments

Plants are frequently attacked by diverse pathogens/pests, and many of them had been reported to trigger a plant mediated “cry for help” phenomenon which enriches disease suppression and growth promotion microbiota in the rhizosphere. Understanding how do plants cry for help to the microbiota is of giant application potential and increasing general interest. The manuscript by Liu et al., elegantly utilized a batch of pathogen mutants, which was believed to trigger different levels of biotic stresses to the host plant. The most important finding here is that infection by avirulent mutant of pathogen could still trigger the occurrence of “disease suppressive soil” without serious disease symptoms. This finding significantly furthered our understanding about the “cry for help” phenomenon, and has broad application potential in agricultural. The authors also characterized the key microbes recruited by upground pathogen infection, and further identified the key metabolic changes in the root exudate which might be related to the recruitment of disease suppressive microbes. Overall, the manuscript is carefully designed and well written, and would be of great interest to the broad readers in plant-microbe interaction field, and fit NC journal.

Response: Thank you for your positive comment. We have revised the manuscript according to your comment. Please find the detailed responses below.

I still have several comments/concerns related to this work:

1) The term “vaccine” usually refers to dead bacterial or virus particles. Can the authors update the term to “attenuated live vaccine” or something like that?

Response: Thank you for your comment. In agreement with you and another reviewer's comments, the manuscript has been revised to avoid using “vaccine”, “antibody” and other words or sentences that humanize plant process.

2)the authors infiltrated “leaves of 15-day-old gnotobiotic Arabidopsis grown in MS medium with Pst DC3000 and the derived mutants”. The leaves of 15-day-old gnotobiotic Arabidopsis should be very tiny and how can this be “infiltrated”? can we get a bit more description in the method. Also, for this system, is there a possibility that pathogen contamination in the rhizosphere, and they would utilize root exudate for proliferation. That will indirectly affect the composition of root exudates?

Response: Thank you for your comment. We are sorry for the mistake, the plant used for infiltration are 4-week-old seedlings. It has been corrected in the revised manuscript. Second, we have checked the contamination by search the microbiome raw data for the 16S rDNA sequence of DC3000, and no hits were obtained. It indicates no DC3000 contamination in the rhizosphere. In addition, we have added more detailed description for the inoculation in the method.

3) the authors systematically studied the changes in the root exudates upon infection, which is a critical part of the manuscript. However, it is not well concluded in the abstract part.

Response: Thank you for your comment. We have concluded the effect of root exudation of myristic acid on “cry for help” response in Abstract.

4)The authors did network analysis for the rhizosphere microbial community changes upon pathogen (mutants) infection (Fig. S6). However, more network related parameters (connectance, module numbers, degree centralization.....) could be listed to show the overall features of networks.

Response: Thank you for your comment. In agreement with another reviewer’s comment that the network analysis did not show considerable support to the main aim of this study, we decide to remove that result to be more focusing in the revised manuscript.

5)In the introduction part (line 65), a lot recent exciting breakthrough which supports that root exudate could shape rhizosphere microbiota should be briefly introduced here (following the root exudate part). Like : Yu, Peng, et al. "Plant flavones enrich rhizosphere Oxalobacteraceae to improve maize performance under nitrogen deprivation." Nature plants 7.4 (2021): 481-499.; Huang A C, Jiang T, Liu Y X, et al. A specialized metabolic network selectively modulates Arabidopsis root microbiota[J]. Science, 2019, 364(6440): eaau6389.

Response: Thank you for your comment. We have carefully revised the introduction to include these important papers.

Reviewers' Comments:

Reviewer #1:

Remarks to the Author:

Thank you for addressing some of my previous comments. I appreciate the additional experiments and findings included in the revised manuscript. However, I find that some key experiments that are required to conclude are still missing.

(Major comments)

1. Regarding my previous major comment 2, it is crucial to understand the underlying factors that lead to the "cry for help" response. While the mutant D36EFLC does not produce the flagellin (therefore flg22), D36EFLC still possesses other MAMPs capable of activating PTI responses. Thus, although flg22-triggered responses are not required for cry for help mediated by D36EFLC, it is very possible that PTI plays a vital role for cry for help. It certainly is of great importance to determine whether PTI is responsible for this. Consequently, I suggest that the application of exogenous flg22 and heat-killed bacteria remains the most practical approach for investigating its specific role in mediating the "cry-for-help" response in this study. Otherwise, it remains unclear what triggers cry for help and at this stage, it can only say that nonpathogenic strains somehow do this.

2. D36E and D36EFLC-affected metabolites alone could induce the plant "cry-for-help" response, but this induction was quenched by live D36EHPM, implying a repressive effect of ETI on cry for help. However, it cannot be determined whether ETI or HopM1 did this when only one ETI-triggering strain was used. Further clarification of this mechanism would significantly contribute to our understanding of the interaction between HopM1, ETI, and the "cry-for-help" response. The authors need to use an additional ETI-triggering strain (another effector that triggers ETI) to determine whether suppression of cry for help was mediated by ETI or HopM1 effector function.

(Other comments)

1. In Fig 3C, why the dots(data) and letters changed compared with the original manuscript?

2. In this manuscript, the authors utilized mutants and derivatives interchangeably. Please maintain consistency throughout the study. Considering you used D36EHPM, derivatives may be better.

Reviewer #2:

Remarks to the Author:

The authors have done a big effort addressing the concerns of all reviewers. On my side, they have justified comments I made and have improved where suggested, and I only have some very minor additional points.

Line 92: tomato should not be in italics

Figure 2: for clarity, you could use the same color code in panels C and D as in A, since you're highlighting the same information as is already shown in A.

Lines 158-160: was there any disease symptoms?

Figure 3: it's confusing that in the text you describe the treatments with some names (e.g. SL-D36EHPM) and in the figures you don't use them.

Lines 167, 253: amplicon sequencing

Figure 4c: jitter seems to be expanded?

Line 202: figure 4?

Reviewer #3:

Remarks to the Author:

As one of the reviewers who previously evaluated the original version of this manuscript, I read this revised version with pleasure. The authors did a very good job in incorporating all issues raised in my previous review report. I very much like the fact that the authors now use the term "nonpathogenic" as opposed to "avirulent", which takes away a lot of unnecessary confusion and as a result it makes the manuscript much better readable. The story is now much cleaner and the take home message that also nonpathogenic variants of phytopathogens can induce the cry for help response and create a soil-borne legacy that protects a next generation of plants is a nice

new chapter to this field of research. I only have a few minor comments:

1. I think the title would read better if it would read "Nonpathogenic mutants of a phytopathogen....". So add "a" before "phytopathogen".
2. Line 34: "elicit different levels" instead of "level".
3. Line 40: ETI: shouldn't abbreviations be given in full at first time use?
4. Line 55: Maybe good to cite here also the first primary papers demonstrating the cry for help response induced by foliar pathogens (Yuan et al. 2018 Microbiome; Berendsen et al. 2018 ISME J.)
5. Line 60: Maybe also good to cite here the Arabidopsis-Hyaloperonospora arabidopsidis pathosystem (Berendsen et al., 2018 ISME J)
6. Line 67-68: Maybe good to cite here also Vismans et al. 2022; Scientific Reports: 12: 22473.
7. Line 92: Tomato should not be written in italics.

Reviewer #4:

Remarks to the Author:

High organisms are in constant association with diverse microbial communities, and the mutual interactions between host and the microbiomes are critical for the fitness of the host. It had been widely observed that plants could enrich beneficial microbes in responses to pathogens, drought and nutrient deprivation stresses. This leads to the "cry for help" hypothesis that host might be able to positively enrich beneficial microbes. Although this phenomenon is of direct application potential in Agriculture, we currently know very little about the biochemical and genetic mechanisms underlying this. The manuscript by Liu et al. firstly utilized genetically modified model pathogen system to study this question, and provided novel and mechanistic insights into the link between bacterial virulence and "cry for help" phenomenon. The authors thoroughly addressed my previous comments, and further provided more mechanistic studies into this phenomenon.

There are at least two novelties from this study: 1. Most "cry for help" system refers to plant mediated enrichment of beneficial microbes in response to soil borne pathogens (wilt, Fusarium, nematodes). However, this study focused on the induction of "disease suppressive soil" by leaf pathogens. Investigating the effect and mechanisms underlying leaf pathogens triggered "cry for help" to root microbiome is a novel field. This also would guide further application of leaf pathogen (or weakened pathogen) to effectively induce disease suppressiveness in soil. I believe this is one interesting and unique point of this study and could be stated in the introduction. 2. The authors found that even low dose of immune activation (induced by DC3000 mutants with weakened virulence) could still enrich beneficial microbes and induce disease suppressive and growth promoting soils over generations. This is the first study show that low dose of immune responses could elicit similar "cry for help" responsiveness over several generations. It suggests the robustness of "cry for help" pathway in plants over long term plant-pathogen-microbiome co-evolution, and has direct application potential in artificially inducing "disease suppressive soil" or alleviating succession cropping obstacle.

Questions/comments:

The authors "checked the virulence of the strains by Arabidopsis leaf infiltration, and found that only wild-type Pst DC3000 (WT) caused disease symptoms, while all other mutants (D36E, D36EFLC and D36EHPM) did not". However, the standard way to check virulence is just plating the pathogen levels (CFU counts). I guess this could also help address previous R3's confusing about virulence stuff.

Recent breakthroughs in plant immune signaling had confirmed the largely shared downstream signaling components and signaling responses between ETI and PTI responses. The mutual interactions between PTI and ETI indicates that ETI might function through reinforcing PTI pathways. It would be hard to understand if PTI could induce "cry for help" while ETI could "quench" this response. The observation that ETI activating in each generation blocks "cry for help" is very cool and interesting. The current data is solid enough. I just wonder that an alternative explanation could be that infection of live DC3000HPM triggers plant stunting (Fig 5c), which might also affect photosynthetic carbon production and root exudation. This might also be

discussed.

Line 65-67, the sentence is too long and complicate. Immunity, hormones, signal transduction are totally different terms and couldn't be listed together here.

Line 72: prevents should be restricts.

Line 125: specific could be essential

D36EHPM treated may block the effect of D36E and D36EFLC on the legume? microbiome.

Response to reviewers' comments

Reviewer #1 (Remarks to the Author):

Thank you for addressing some of my previous comments. I appreciate the additional experiments and findings included in the revised manuscript. However, I find that some key experiments that are required to conclude are still missing.

Response: Thank you for your overall comment. Please find the detailed point-by-point response below.

(Major comments)

1. Regarding my previous major comment 2, it is crucial to understand the underlying factors that lead to the "cry for help" response. While the mutant D36EFLC does not produce the flagellin (therefore flg22), D36EFLC still possesses other MAMPs capable of activating PTI responses. Thus, although flg22-triggered responses are not required for cry for help mediated by D36EFLC, it is very possible that PTI plays a vital role for cry for help. It certainly is of great importance to determine whether PTI is responsible for this. Consequently, I suggest that the application of exogenous flg22 and heat-killed bacteria remains the most practical approach for investigating its specific role in mediating the "cry-for-help" response in this study. Otherwise, it remains unclear what triggers cry for help and at this stage, it can only say that nonpathogenic strains somehow do this.

Response: Thank you for your comments. Following your suggestions, we have performed a three-generation enrichment of the soil-borne legacy to test whether flg22 and heat-killed DC3000 could triggered a soil-borne legacy with beneficial effects. The results showed that flg22 alone or the heat-killed DC3000 could trigger a "cry for help" response. Briefly, Arabidopsis grown in SL-flg22 or SL-heat-killed DC3000 showed significantly higher fresh weigh, germination ratio and reduced infection by DC3000, although in a less extent as that in the SL-D36EFLC/S (soil containing the soil-borne legacy from D36EFLC supernatant-treated plants). This result indicated that PTI is sufficient to trigger the "cry for help" response.

2. D36E and D36EFLC-affected metabolites alone could induce the plant "cry-for-help" response, but this induction was quenched by live D36EHPM, implying a repressive effect of ETI on cry for help. However, it cannot be determined whether ETI or HopM1 did this when only one ETI-triggering strain was used. Further clarification of this mechanism would significantly contribute to our understanding of the interaction between HopM1, ETI, and the "cry-for-help" response. The authors need to use an additional ETI-triggering strain (another effector that triggers ETI) to determine whether suppression of cry for help was mediated by ETI or HopM1 effector function.

Response: Thank you for your comments. Following your suggestions, we further tested another ETI-triggering strain, D36EavrRpt2, to determine whether suppression of "cry for help" was mediated by ETI or specifically by HopM1. Interestingly, the beneficial effect of D36EFLC/S on soil-borne legacy was quenched by D36EavrRpt2 as well as D36EHPM, indicating the suppression of "cry for help" response is mediated by ETI, rather than the HopM1 effector function.

(Other comments)

1. In Fig 3C, why the dots(data) and letters changed compared with the original manuscript?

Response: Sorry for the mistakes. We made a mistake in copy and paste of the data from excel to plotting and statistical software in the first version, and we found and corrected this mistake during replotting the figure in the first round of revision.

2. In this manuscript, the authors utilized mutants and derivatives interchangeably. Please maintain consistency throughout the study. Considering you used D36EHPM, derivatives may be better.

Response: Thank you for your comment. We have revised “mutants” to “derivatives” throughout the manuscript.

Reviewer #2 (Remarks to the Author):

The authors have done a big effort addressing the concerns of all reviewers. On my side, they have justified comments I made and have improved where suggested, and I only have some very minor additional points.

Response: Thank you for your positive comment.

Line 92: tomato should not be in italics

Response: Thank you for your comment. The text has been revised accordingly.

Figure 2: for clarity, you could use the same color code in panels C and D as in A, since you're highlighting the same information as is already shown in A.

Response: Thank you for your comment. The figure color has been revised accordingly.

Lines 158-160: was there any disease symptoms?

Response: Thank you for your comment. The disease symptoms have been described.

Figure 3: it's confusing that in the text you describe the treatments with some names (e.g. SL-D36EHPM) and in the figures you don't use them.

Response: Thank you for your comment. We have revised the names in the figures according to the text.

Lines 167, 253: amplicon sequencing

Response: Thank you for your comment. The text has been revised accordingly.

Figure 4c: jitter seems to be expanded?

Response: Thank you for your comment. The figure panel has been revised to its original ratio.

Line 202: figure 4?

Response: Sorry for the mistake. It's figure 4.

Reviewer #3 (Remarks to the Author):

As one of the reviewers who previously evaluated the original version of this manuscript, I read this revised version with pleasure. The authors did a very good job in incorporating all issues raised in my previous review report. I very much like the fact that the authors now use the term "nonpathogenic" as opposed to "avirulent", which takes away a lot of unnecessary confusion and as a result it makes the manuscript much better readable. The story is now much cleaner and the take home message that also nonpathogenic variants of phytopathogens can induce the cry for help response and create a soil-borne legacy that protects a next generation of plants is a nice new chapter to this field of research. I only have a few minor comments:

Response: Thank you for your positive comments.

1. I think the title would read better if it would read "Nonpathogenic mutants of a phytopathogen....". So add "a" before "phytopathogen".

Response: Thank you for your comment. We have revised the title according to your comment.

2. Line 34: "elicit different levels" instead of "level".

Response: Thank you for your comment. We have revised the word.

3. Line 40: ETI: shouldn't abbreviations be given in full at first time use?

Response: Sorry for the mistake. We have typed the full name of ETI in abstract.

4. Line 55: Maybe good to cite here also the first primary papers demonstrating the cry for help response induced by foliar pathogens (Yuan et al. 2018 Microbiome; Berendsen et al. 2018 ISME J.)

Response: Thank you for your comment. The first primary papers demonstrating the cry for help response induced by foliar pathogens has been cited here.

5. Line 60: Maybe also good to cite here the Arabidopsis-Hyaloperonospora arabidopsidis pathosystem (Berendsen et al., 2018 ISME J)

Response: Thank you for your comment. This pathosystem has been cited in the text accordingly.

6. Line 67-68: Maybe good to cite here also Vismans et al. 2022; Scientific Reports: 12: 22473.

Response: Thank you for your comment. This paper has been cited accordingly.

7. Line 92: Tomato should not be written in italics.

Response: Thank you for your comment. The text has been revised.

Reviewer #4 (Remarks to the Author):

High organisms are in constant association with diverse microbial communities, and the mutual interactions between host and the microbiomes are critical for the fitness of the host. It had been widely observed that plants could enrich beneficial microbes in responses to pathogens, drought and nutrient deprivation stresses. This leads to the “cry for help” hypothesis that host might be able to positively enrich beneficial microbes. Although this phenomenon is of direct application potential in Agriculture, we currently know very little about the biochemical and genetic mechanisms underlying this. The manuscript by Liu et al. firstly utilized genetically modified model pathogen system to study this question, and provided novel and mechanistic insights into the link between bacterial virulence and “cry for help” phenomenon. The authors thoroughly addressed my previous comments, and further provided more mechanistic studies into this phenomenon.

There are at least two novelties from this study: 1. Most “cry for help” system refers to plant mediated enrichment of beneficial microbes in response to soil borne pathogens (wilt, Fusarium, nematodes). However, this study focused on the induction of “disease suppressive soil” by leaf pathogens. Investigating the effect and mechanisms underlying leaf pathogens triggered “cry for help” to root microbiome is a novel field. This also would guide further application of leaf pathogen (or weakened pathogen) to effectively induce disease suppressiveness in soil. I believe this is one interesting and unique point of this study and could be stated in the introduction. 2. The authors found that even low dose of immune activation (induced by DC3000 mutants with weakened virulence) could still enrich beneficial microbes and induce disease suppressive and growth promoting soils over generations. This is the first study show that low dose of immune responses could elicit similar “cry for help” responsiveness over several generations. It suggests the robustness of “cry for help” pathway in plants over long term plant-pathogen-microbiome co-evolution, and has direct application potential in artificially inducing “disease suppressive soil” or alleviating succession cropping obstacle.

Questions/comments:

The authors “checked the virulence of the strains by Arabidopsis leaf infiltration, and found that only wild-type Pst DC3000 (WT) caused disease symptoms, while all other mutants (D36E, D36EFLC and D36EHPM) did not”. However, the standard way to check virulence is just plating the pathogen levels (CFU counts). I guess this could also help address previous R3’s confusing about virulence stuff.

Response: Thank you for your comment. This experiment has been reperformed by CFU counting according to your comment. Please find the data shown in Fig. 1C.

Recent breakthroughs in plant immune signaling had confirmed the largely shared downstream signaling components and signaling responses between ETI and PTI responses. The mutual interactions between PTI and ETI indicates that ETI might function through reinforcing PTI pathways. It would be hard to understand if PTI could induce “cry for help” while ETI could “quench” this response. The observation that ETI activating in each generation blocks “cry for help” is very cool and interesting. The current data is solid enough. I just wonder that an alternative explanation could be that infection of live DC3000HPM triggers plant stunting (Fig

5c), which might also affect photosynthetic carbon production and root exudation. This might also be discussed.

Response: Thank you for your comment. We agree with your opinion that our results cannot exclude the possibility that ETI caused plant stunting to contribute to this phenomenon. We have added a short discussion in the last to describe this explanation.

Line 65-67, the sentence is too long and complicate. Immunity, hormones, signal transduction are totally different terms and couldn't be listed together here.

Response: Thank you for your comment. We have removed “hormone production” and “signaling transduction” here.

Line 72: prevents should be restricts.

Line 125: specific could be essential

D36EHPM treated may block the effect of D36E and D36EFLC on the legume? microbiome.

Response: Thank you for your comment. The text has been revised accordingly.

Reviewers' Comments:

Reviewer #1:

Remarks to the Author:

I appreciate the authors' effort to address the important question of whether PTI activation can trigger cry for help. Accordingly, the authors have professionally addressed my previous comments. Nevertheless, I have several suggestions to further improve this excellent manuscript. The discovery that flg22 and heat-killed DC3000-triggered PTI can induce the "cry for help" is a very important finding. Therefore, this should be mentioned in Summary.

Concerning line 330, the authors would need supporting evidence that flg22 alone significantly impacts root exudates. I suggest re-wording to reflect the reality.

The order of Fig2 is messed up. Thus, some descriptions are not consistent with the order of figures. (eg. Line 135) Please check.

The term "Arabidopsis" is sometimes italicized. Either italic or not should be consistent.

Line 149, "s" in "soil" was underlined. If the authors want to emphasize the SL in the following text, "L" should be underlined, too.

Line 251 and 273, "l" in "legacy" was underlined.

In Supplemental Figure S3, most of the "D36EFLC" was written by "D36ELC". These should be corrected.

Reviewer #4:

Remarks to the Author:

I read through the manuscript and I believe the authors fixed the major point 1 and point 2 concerns.

The authors conducted new experiments using exogenous flg22 and heat-killed bacteria to confirm the PTI activation in leaves is sufficient to trigger "cry for help". This deepens our understanding about what exactly is the "cry for help" signal.

Meanwhile, the authors further confirms that both avrRpt2 and HopM1 triggered ETI cannot trigger "cry for help" signal.

Overall, this work thoroughly investigated effects of PTI and ETI on "cry for help" phenotype, which provided valuable and fundamental insights underlying the "cry for help" hypothesis.

Response to reviewers' comments

Reviewer #1 (Remarks to the Author):

I appreciate the authors' effort to address the important question of whether PTI activation can trigger cry for help. Accordingly, the authors have professionally addressed my previous comments. Nevertheless, I have several suggestions to further improve this excellent manuscript.

The discovery that flg22 and heat-killed DC3000-triggered PTI can induce the "cry for help" is a very important finding. Therefore, this should be mentioned in Summary.

Response: Thank you for your suggestion. We have mentioned this finding in the abstract.

Concerning line 330, the authors would need supporting evidence that flg22 alone significantly impacts root exudates. I suggest re-wording to reflect the reality.

Response: Sorry for the mistake. We have revised the sentence to reflect the reality.

The order of Fig2 is messed up. Thus, some descriptions are not consistent with the order of figures. (eg. Line 135) Please check.

Response: Thank you for your comment. The order of Fig. 2 has been revised.

The term "Arabidopsis" is sometimes italicized. Either italic or not should be consistent.

Response: Thank you for your comment. All "Arabidopsis" have been revised to be consistent.

Line 149, "s" in "soil" was underlined. If the authors want to emphasize the SL in the following text, "L" should be underlined, too.

Line 251 and 273, "l" in "legacy" was underlined.

Response: Thank you for your comment. We have checked all the letters underlined.

In Supplemental Figure S3, most of the "D36EFLC" was written by "D36ELC". These should be corrected.

Response: Thank you for your comment. The wrong label has been revised to "D36EFLC" in the figure.

Reviewer #4 (Remarks to the Author):

I read through the manuscript and I believe the authors fixed the major point 1 and point 2 concerns. The authors conducted new experiments using exogenous flg22 and heat-killed bacteria to confirm the PTI activation in leaves is sufficient to trigger "cry for help". This deepens our understanding about what exactly is the "cry for help" signal.

Meanwhile, the authors further confirm that both avrRpt2 and HopM1 triggered ETI cannot trigger "cry for help" signal.

Overall, this work thoroughly investigated effects of PTI and ETI on "cry for help" phenotype, which provided valuable and fundamental insights underlying the "cry for help" hypothesis.

Response: Thank you for your positive comment.